# PRISM-PHYSICS: CAUSAL DAG-BASED PROCESS EVALUATION FOR PHYSICS REASONING

**Wanjia Zhao**[*1] **Qinwei Ma**[*2] **Jingzhe Shi**[*2] **Shirley Wu**[1] **Jiaqi Han**[1] **Yijia Xiao**[3]
**Si-Yuan Chen**[4] **Xiao Luo**[5] **Ludwig Schmidt**[1] **James Zou**[1]
[1]Stanford University [2]Tsinghua University [3]University of California Los Angeles
[4]Harvard University [5]University of Wisconsin–Madison

## ABSTRACT

Benchmarks for competition-style reasoning have advanced evaluation in mathematics and programming, yet physics remains comparatively explored. Most existing physics benchmarks evaluate only final answers, which fail to capture reasoning processes, while recent stepwise methods rely on heuristic LLM-as-judge scoring or restrictive linear assumptions, limiting reliability and diagnostic validity. We introduce PRISM-PHYSICS, a process-level evaluation framework and benchmark for complex physics reasoning problems. Solutions are represented as directed acyclic graphs (DAGs) of formulas, explicitly encoding causal dependencies among intermediate steps to enable fine-grained, interpretable, and theoretically grounded scoring. We prove the optimality of the DAG representation and the corresponding scoring policy. Combining with a fully rule-based method for symbolic formula equivalence matching that we developed, we ensure consistent validation across diverse formulations without heuristic judgments. Results show that our evaluation framework is more aligned with human experts' scoring. Experiments on state-of-the-art LLMs reveal persistent reasoning failures in physics, while step-level scoring offers both diagnostic insight and rich signals for later training. By combining structural rigor, theoretical guarantees, and symbolic validation, PRISM-PHYSICS provides a principled foundation for advancing process-level evaluation and guiding the development of models with deeper scientific reasoning capabilities.

**Project Page:** https://open-prism.github.io/PRISM-Physics/

## 1 INTRODUCTION

Benchmarks for competition-style reasoning have advanced rapidly in mathematics (e.g., IMO) (Zheng et al., 2021; He et al., 2024; Gao et al., 2024) and programming (e.g., IOI) (Shi et al., 2024; Zhu et al., 2025; El-Kishky et al., 2025), providing comprehensive testbeds for evaluating large language models (LLMs). In contrast, physics competitions remain comparatively underserved, despite requiring not only deep domain knowledge, but also advanced analytical modeling, multi-step symbolic derivation, and precise numerical computation. These skills are fundamental indicators of scientific reasoning ability, as they integrate conceptual understanding, modeling assumptions, and rigorous problem-solving under complex constraint (Jaiswal et al., 2024). Consequently, developing fine-grained evaluation frameworks and benchmarks for competition-level physics is essential for systematically assessing and advancing LLMs' capabilities in this critical domain (Chang et al., 2024; Song et al., 2025).

A major obstacle is that physics derivations are inherently non-linear: solutions frequently branch, merge, or reuse intermediate results. Existing benchmarks fail to capture this structure due to three key limitations: **1.** Most physics benchmarks rely on multiple-choice or short-answer formats (Wang et al.; Rein et al., 2024), evaluating only the final answer and ignoring the reasoning process. Such formats obscure the reasoning process and provide limited diagnostic value for understanding model capabilities. **2.** Most also rely on LLM-as-judge scoring (He et al., 2024; Xiang et al., 2025), which is prone to hallucinations, prompt sensitivity, and inconsistent grading. **3.** While some recent

---

*Equal contribution. Correspondence to wanjiazh@cs.stanford.edu

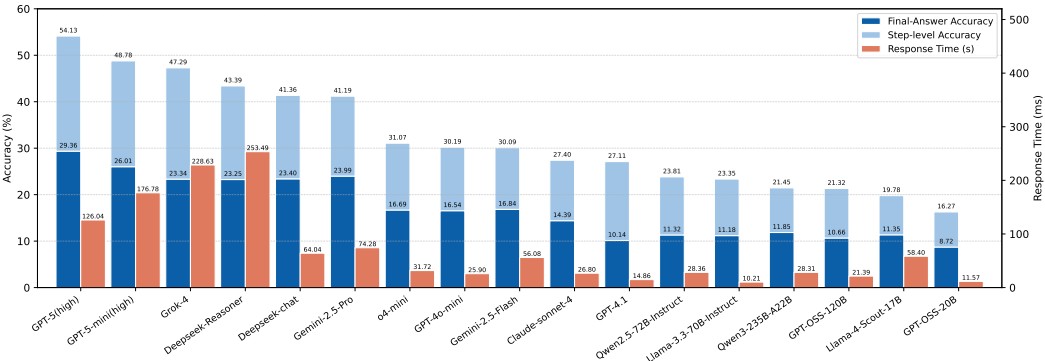

Figure 1: Model performance on PRISM-Physics. We reported both Final-Answer Accuracy, Step-level Accuracy and Response Time.

work (Zhang et al., 2025) has made initial attempts at step-by-step scoring, these approaches often rely on strong assumptions such as strictly linear step ordering or shallow expression matching that limit the validity and generalizability of the framework. As a result, current methodologies remain inadequate for revealing the systematic reasoning failures of LLMs in physics problem solving.

To address these limitations, we introduce **PRISM-Physics**, a process-level evaluation framework and benchmark that represents physics solutions as directed acyclic graphs (DAGs) of formulas. This graph-based structure explicitly encodes causal dependencies among intermediate steps, enabling fine-grained scoring that is theoretically grounded and interpretable. To ensure consistency, we further develop a fully rule-based symbolic equivalence checker, which provides robust validation across diverse formulations and removes dependence on heuristic judgments.

Our main contributions are summarized as follows: **1.** We construct a large-scale benchmark of competition-level physics problems with carefully curated, DAG-structured solutions. **2.** We propose a *DAG-based scoring policy* that explicitly models causal dependencies among formulas, enabling fine-grained and interpretable process-level evaluation. We further provide a theoretical proof of its optimality, showing that it minimizes evaluation ambiguity and aligns naturally with the logical structure of physics derivations. **3.** We develop a fully rule-based *symbolic formula equivalence checker* to reliably validate diverse mathematical expressions, ensuring consistent comparison across alternative formulations and eliminating reliance on heuristic LLM-as-judge scoring. **4.** We conduct extensive experiments on a broad range of LLMs, revealing persistent challenges in sustaining coherent reasoning chains and in correctly applying physical principles. Furthermore, we systematically compare our evaluation framework with existing approaches, demonstrating its *superior reliability, interpretability, and diagnostic power* for evaluating process-level reasoning capabilities.

Taken together, PRISM-Physics establishes the first principled foundation for process-level evaluation in physics, bridging structural rigor, theoretical guarantees, and symbolic validation.

## 2 RELATED WORK

**Physics Benchmark.** Physics problems, as a proxy of how LLMs understand Physics, have been used as benchmarks for LLMs in recent years. Especially, previous work has been using Physics Olympiad problems for benchmarking LLM reasoning and problem solving abilities. For instance, OlympiadBench (He et al., 2024) aggregates problems from multiple Olympiads; SeePhys (Xiang et al., 2025) incorporates visual problems to study how visual ability improves LLM performance; PhyBench (Qiu et al., 2025) focuses on rigor and originality. While such benchmarks propose metrics (e.g., EED score (Qiu et al., 2025)), they still focus primarily on final answers and fail to represent or provide more fine-grained process scores. More recently, process-based evaluation of LLM reasoning has become a focus. PhysReason (Zhang et al., 2025) evaluates intermediate steps by checking correctness of expressions and assigning linear scores, but this approach is restricted to expressions (rather than equations) and cannot represent the dependency logic among steps.

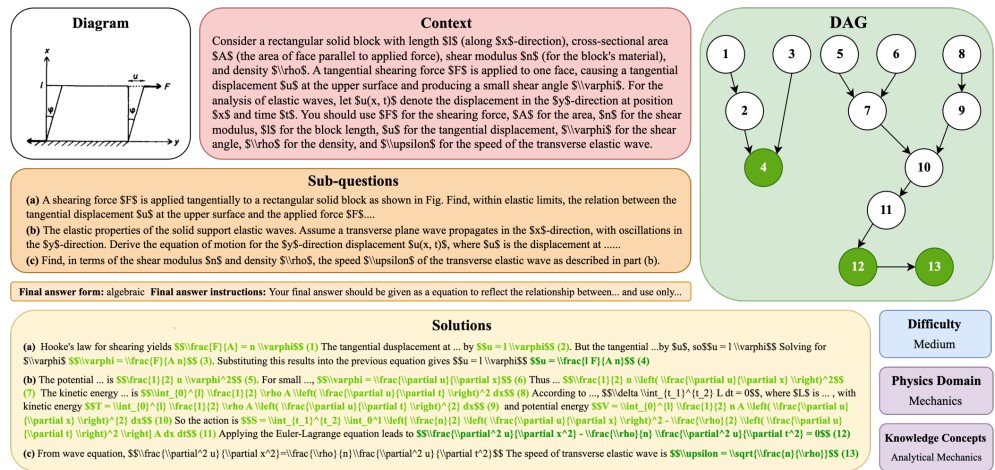

Figure 2: A data example with the proposed DAG structure.

**LLM-as-Judge for Problem Solving.** Reliable evaluation of physics problem solving requires assessing not only final answer correctness, but also the validity of intermediate reasoning steps. Human expert annotation, while generally reliable, is costly and unscalable in large-scale (Gu et al., 2024; Liu et al., 2023; Ye et al., 2025; Petrov et al., 2025; Mao et al., 2024). Automated LLM-as-judge methods have shown potential in mathematical and physics tasks, but are still susceptible to errors from implicit assumptions, symbolic manipulation errors, and misinterpretation of domain concepts (Zheng et al., 2023; Gu et al., 2024; Ye et al., 2025; Hendrycks et al., 2021; Gulati et al., 2024; Liu et al., 2024; Lu et al., 2024). This challenge is amplified in physics, where various physical concepts, constants, and equivalent formulations create many valid variations of the same expression, making judgment more difficult (Wang et al.; Xu et al., 2025; Zhang et al., 2025).

To address these issues, we introduce a formula-based verification framework that directly compares symbolic expressions for physical and mathematical consistency, offering a faster and more reliable alternative to costly human annotation (Brence et al., 2023; Gao et al., 2023; Chen et al., 2023; Cobbe et al., 2021; Hendrycks et al., 2021; Xia et al., 2024; Gao et al., 2025; Li et al., 2024).

## 3 PRELIMINARY AND FORMULATION

### 3.1 FORMULA-MATCHING-BASED EVALUATION

A crucial component that distinguishes our evaluation framework from most LLM-as-judge counterparts is that, ours is purely based on rule-based **Formula Equivalence Matching**, also called as **Formula Matching**. More specifically, we say two formulas are **matched** if they are mathematically equivalent. In Section 4.1 we will show how we actually match two formulas, and discuss our improvement against previous work.

### 3.2 MOTIVATION FOR THE DAG STRUCTURE

Naive process-scoring policies present inherent limitations: *strict matching* fails to recognize correct outcomes obtained through alternative derivations, whereas *prefix credit* overestimates performance by indiscriminately assigning credit to all prior steps once a single formula is matched. To address these issues, we represent the reference solution as a directed acyclic graph (DAG) of formulas, where the edges encode explicit prerequisite relations. With this structure, credit propagates only along causal chains from matched nodes to their ancestors. This approach (i) avoids the harshness of strict matching by rewarding justified intermediate work, (ii) prevents the over-crediting of prefix policies by restricting propagation to prerequisites only, and (iii) offers a representation intuitive for human reasoning, reliable for LLM annotation, and theoretically complete under mild assumptions.

### 3.3 DAG REPRESENTATION OF SOLUTIONS.

Formally, a *directed acyclic graph* (DAG) is a pair $G = (V, E)$, where $V$ is a finite set of nodes and $E \subseteq V \times V$ is a set of directed edges such that there is no directed cycle in $G$. That is, there is no sequence of distinct nodes $v_1, v_2, \ldots, v_k$ with $k \geq 2$ satisfying $(v_i, v_{i+1}) \in E$ for all $1 \leq i < k$ and $(v_k, v_1) \in E$. A DAG thus encodes a partial order over its nodes, which is particularly suitable for representing stepwise logical or computational dependencies.

In our setting, each solution is systematically converted into such a DAG: **1. Nodes (formulas).** Each $v \in V$ denotes a canonicalized LaTeX expression representing a mathematically key step (e.g., physical law, derived intermediate equation, simplified relation). Canonicalization guarantees syntactic and semantic consistency across solutions. **2. Edges (dependencies).** For $(u, v) \in E$, formula $v$ is derived from formula $u$. By construction, the edges reference only prior nodes, thereby ensuring temporal causal consistency and a valid topological ordering aligned with natural reasoning. **3. Minimality.** Redundant algebraic steps are removed, retaining only essential formulas, thereby yielding a concise yet sufficient structure that captures the core reasoning trajectory. **4. Completeness.** Every node must be connected by a directed path to at least one designated *final answer node*. Thus, all preserved formulas contribute causally to the derivation of the final solution, ensuring that there are no dangling or irrelevant steps.

The resulting DAG captures the **logical skeleton of the solution**, where nodes formalize reasoning steps and edges encode causal dependencies. In this way, the derivation is made machine-interpretable, with correctness evaluable both locally (per node) and globally (via entire dependency chains). This structure provides the foundation for our scoring mechanics (see Section 3.4).

### 3.4 ANCESTOR CLOSURE SCORING

**Definition 1** (Ancestor Closure). *Let* $\mathcal{M} \subseteq \mathcal{F}$ *be the set of matched reference formulas, and let* $\mathrm{Anc}(S) := \{A \in \mathcal{F}/\mathcal{S} : \exists B \in S \text{ and a path } A \prec \cdots \prec B\}$. *We define* ancestor closure *of* $\mathcal{M}$ *as:*

$$\mathrm{Ach}(\mathcal{M}) := \mathcal{M} \cup \mathrm{Anc}(\mathcal{M}),$$

*i.e., all matched nodes and all their DAG ancestors (reverse reachability).*

By this definition, we formally propose our **Ancestor Closure Scoring Policy**:

**Definition 2** (Ancestor Closure Scoring Policy). *Let* $\mathcal{M}$ *be the set of formulas in the reference DAG that are directly matched by the submission (student solution), then **Ancestor Closure Scoring** gives the final score as*

$$S = \frac{|\mathrm{Ach}(\mathcal{M})|}{|\mathcal{F}|}, \tag{1}$$

*where* $\mathcal{F}$ *is the set of all formulas in the DAG.*

*Intuition.* In a solution DAG, the edges point from the prerequisite formulas to their dependents. If a dependent formula is achieved (matched), then every formula that lies on any directed path *into* it is considered achieved as well, because those predecessors are logically required for deriving it. Thus, we count the union of each matched formula together with all of its prerequisites; the score is simply the fraction of reference formulas covered by this union.

### 3.5 JUSTIFICATION SYSTEM AND OPTIMALITY OF DAG

We formulate a good formula-based scoring policy with the following intuition: a scoring policy should first verify which formulas in the reference solution are *matched* by the student solution, then see if some other formulas can be *justified* by the matched reference formulas. We restrict attention to complete justifications; partial or approximate justifications are outside the scope of this formulation. Under such a formulation, we can formally discuss the optimality of our provided DAG structure as a representation of the formula relations, as well as our scoring policy.

**Definition 3** (Justification System). *Let* $\mathcal{F}$ *be the set of reference formulas. A* justification relation *is a relation* $\Rightarrow \subseteq 2^{\mathcal{F}} \times \mathcal{F}$, *where* $X \Rightarrow B$ *means: once every formula in* $X$ *is matched, the formula* $B$ *is automatically warranted, or in other words, adding formula* $B$ *into the student solution will*

*not make any further progress to the final answer. The set $F$ and all justification relations within it form the* justification system $(F, \Rightarrow)$.

*We define the* minimal justification relation $\vdash$ *by*

$$A \vdash B \quad \Longleftrightarrow \quad A \Rightarrow B \text{ and no proper subset } Y \subset A \text{ satisfies } Y \Rightarrow B.$$

*In this case, $A$ is called a* minimal justifier *of $B$.*

*The* minimal justification kernel *of a justification system $(\mathcal{F}, \Rightarrow)$ is the set*

$$\mathcal{K} = \{(A, B) \in 2^{\mathcal{F}} \times \mathcal{F} : A \vdash B\}.$$

*Two justification systems $(\mathcal{F}, \Rightarrow_1)$ and $(\mathcal{F}, \Rightarrow_2)$ are said to be* equivalent *if they have the same minimal justification kernel, i.e.*

$$\{(A, B) : A \vdash_1 B\} = \{(A, B) : A \vdash_2 B\}.$$

**Assumption 1** (Singleton Minimal Justifiers). *For every $B \in \mathcal{F}$, every minimal justifier of $B$ has size $1$. Equivalently, the justification system can be represented with a binary relation $\prec \subseteq \mathcal{F} \times \mathcal{F}$ such that $A \prec B$ iff $\{A\}$ is a minimal justifier of $B$.*

**Remark.** *Intuitively, $A \prec B$ means $A$ is a* prerequisite *of $B$: if $B$ is awarded, $A$ must also be awarded. Scoring thus flows* upward *in the DAG: from every scored node $B$, all $\prec$-ancestors $A$ are also scored.*

**Assumption 2** (Causality). *If $A \prec B$, then $A$ occurs* earlier *than $B$ in the reference solution's logical order.*

**Theorem 1** (Bijection between order-keeping justifications and DAGs). *Fix $\mathcal{F} = \{F_1, \ldots, F_N\}$ with the index order $1 < \cdots < N$. Let* Just *be the class of justification kernels $\vdash \subseteq \mathcal{F} \times \mathcal{F}$ that satisfy Assumptions 1 and 2 (so $F_i \vdash F_j \Rightarrow i < j$). Let* DAG *be the class of directed acyclic graphs $G = (\mathcal{F}, E)$ whose edges point forward in the index order (i.e., $(F_i, F_j) \in E \Rightarrow i < j$).*

*Define the maps*

$$\Phi : \mathsf{Just} \to \mathsf{DAG}, \qquad \Phi(\vdash) = \big(\mathcal{F}, \ E_{\vdash} := \{(A, B) : A \vdash B\}\big),$$

$$\Psi : \mathsf{DAG} \to \mathsf{Just}, \qquad \Psi\big((\mathcal{F}, E)\big) = \vdash_E \text{ where } A \vdash_E B \iff (A, B) \in E.$$

*Then:*

*(i) $\Phi$ is injective: if $\Phi(\vdash_1) = \Phi(\vdash_2)$, then $\vdash_1 = \vdash_2$.*

*(ii) $\Psi$ is injective: if $\Psi(G_1) = \Psi(G_2)$, then $G_1 = G_2$.*

*Consequently, $\Phi$ and $\Psi$ are mutual inverses and yield a bijection* $\mathsf{Just} \cong \mathsf{DAG}$.

**Remark.** *By Theorem 1, we can see that the DAG representation is precisely the* minimal encoding *of a justification system: it records only the minimal justification kernel and thus contains no redundant rules. At the same time, its closure recovers the full justification system, so the DAG captures exactly the necessary structure with no loss of information and no superfluous complexity. The proof of Theorem 1 is given in Appendix A.1.*

### 3.6 ADMISSIBILITY OF ANCESTOR CLOSURE SCORING

Intuitively, a good scoring policy would map each formula in the DAG to 1 (achieved) or 0 (not achieved), then provide a score accordingly. More formally, we define an admissible scoring policy as follows:

**Definition 4** (Admissible Scoring Policy). *A mapping $\mathsf{S} : 2^{\mathcal{F}} \prec 2^{\mathcal{F}}$, where $\mathsf{S}(\mathcal{M})$ is the set of scored formulas for matched set $\mathcal{M}$, is* admissible *if it satisfies:*
*1.Matched Inclusion: $\mathcal{M} \subseteq \mathsf{S}(\mathcal{M})$.*
*2.Ancestor Closure: If $B \in \mathsf{S}(\mathcal{M})$ and $A \prec B$, then $A \in \mathsf{S}(\mathcal{M})$. ("B justifies A" $\Rightarrow$ back-credit A)*
*3.Soundness: $\mathsf{S}(\mathcal{M}) \subseteq \mathrm{Ach}(\mathcal{M})$. (no over-credit beyond justified ancestors)*

**Theorem 2** (Exact Characterization of Scored Formulas). *For any matched set $\mathcal{M} \subseteq \mathcal{F}$ and any admissible scoring policy $\mathsf{S}$,*

$$\mathsf{S}(\mathcal{M}) = \mathrm{Ach}(\mathcal{M}) = \mathcal{M} \cup \mathrm{Anc}(\mathcal{M}).$$

**Remark.** *Theorem 2 shows our Ancestor Closure Scoring is equivalent to any admissible policy.*

## 4 EVALUATION FRAMEWORK: PRISM-DAG

### 4.1 RULE-BASED PHYSICS FORMULA EQUIVALENCE MATCHING

A key component of our PRISM-DAG is to decide whether two formulas are equivalent. However, checking equivalence between physics formulas presents three key challenges: **(1) Equivalence of equations.** Checking formula equivalence is more difficult than expression comparison; **(2) Constant substitutions.** Two equivalent variables might be written in different forms; **(3) Unit conversion.** Values can be expressed in different units. Prior work often avoids these issues by checking only final expressions, enforcing specific formats, or relying on LLM-as-Judge for comparison, but such approaches either miss process-level evaluation or suffer from hallucination.

To enable fine-grained and rigorous process-based scoring, we propose a two-stage algorithm for physics formula equivalence checking:

**[Stage 1] Constant Substitution**. We substitute certain variables with their expressions. Variables, constants, and units are normalized to predefined form for consistency.

**[Stage 2] Solution Set Equivalence Check**. For two equations with $N$ variables, one variable is randomly chosen as the target, the remaining $N-1$ are assigned random values, then the target is solved to compare whether the solution sets are equivalent. This process is repeated for multiple iterations. Solution set equivalence serves as a proxy for equation equivalence.

Details of the equivalence matching procedure and algorithm are given in Appendix B.1 and Algorithm 1 in Appendix B.2, which successfully resolve these difficulties.

### 4.2 SCORING PIPELINE

Given a student solution and a problem with annotated DAG, we can summarize our evaluation process PRISM-DAG as three steps, details shown in Algorithm 2 in Appendix B.3.

**Formula Extraction and Normalization.** Given a student's solution, all mathematical expressions are first extracted and rewritten into our dataset's standardized canonical form, discarding invalid expressions such as syntactically malformed formulas or irrelevant numerical fragments.

**Formula Matching.** Each standardized student formula is compared against the reference DAG of the solution according to Section 4.1, which outputs a set of matched formulas in the DAG.

**Scoring.** Finally, we score the student solution according to the Ancestor Closure Scoring Policy in Section 3.4 with the DAG and the set of matched formulas.

## 5 PRISM-PHYSICS BENCHMARK

### 5.1 BENCHMARK OVERVIEW

We first collect and preprocess the data to satisfy the need of our scoring pipeline, then do difficulty annotation and domain categorization to give more fine-grained results for analysis. An example of our dataset is shown in Figure 2.

### 5.2 DATA COLLECTION AND PREPROCESSING

**Three-Step Rewriting Pipeline.** To guarantee both internal consistency and external evaluability, every sample in the dataset is processed through a structured three-stage rewriting pipeline. Each stage focuses on eliminating ambiguity and enforcing standardization, while preserving the fidelity of the original content: *(1) Formula normalization.* All mathematical expressions are standardized in LaTeX, with uniform rules for symbolic equivalence and numerical precision; *(2) Context clarification.* Problem statements are rewritten to define all variables and answer requirements explicitly, removing ambiguities; *(3) DAG construction.* Each solution is represented as a directed acyclic graph (DAG) of formulas, verified by rule-based and LLM-based checks.

**Verification and Quality Control.** At each stage, an LLM-based module verifies the formatting, clarity, and dependency rules; failures trigger corrective feedback and regeneration.

**Fine-Grained Enhancements.** Beyond the main pipeline, we applied several refinements: enforcing significant figure rules, explicitly defining all constants and variables, and unifying answer formatting.

See Appendix C for further details and prompts.

### 5.3 ANNOTATION & ANALYSIS

**Difficulty Annotation.** Each problem is assigned a composite difficulty label that integrates LLM-based ratings of conceptual depth and computational burden with an entropy-based DAG complexity measure. The three components are combined into a unified score, which is mapped to *Easy*, *Medium*, *Hard*, capturing both the content difficulty and the reasoning complexity of the solution.

**Physics Domain Categorization.** Each problem is categorized into one of seven key physics domains: (1) *Mechanics*, (2) *Electromagnetism*, (3) *Optics*, (4) *Atomic, Nuclear, and Particle Physics*, (5) *Thermodynamics and Statistical Physics*, (6) *Quantum Mechanics*, and (7) *Solid State Physics and Miscellaneous Topics*. Further details and prompts are provided in Appendix D.

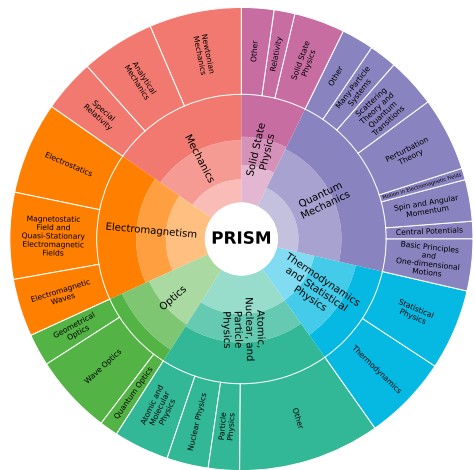

Figure 3: Statistics of PRISM-PHYSICS hierarchical topics and difficulty level.

## 6 EXPERIMENTS

### 6.1 SETTING

We evaluate our proposed PRISM-DAG evaluation framework on the benchmark. We consider two experimental settings: a *text-only* setting, where the problem statement is presented as plain text, and a *multimodal* setting, where relevant diagrams or figures are included alongside the text.

**Models.** We evaluate a diverse set of frontier LLMs. To guide the models in generating reasoning-augmented responses, we design zero-shot COTzheg prompts that encourage step-by-step derivations. All models are run with a unified inference configuration, including fixed temperature, maximum generation length, and identical prompt templates, to ensure fair comparison across settings.

**Evaluation Framework Baselines.** For comparison, we evaluate against: (1) **LLM-as-Judge Scoring**, where an LLM evaluates both the final answer and the solution process given grading prompts(following the evaluation setting of SeePhy (Xiang et al., 2025)); (2) **PSAS-S** (Zhang et al., 2025), an existing process-based framework with strong step-format and ordering assumptions, replicated per its original implementation for fair comparison. Details are provided in Appendix E.

### 6.2 MAIN RESULTS

Table 1 shows Step-level Accuracy and Final-Answer Accuracy across difficulty levels. Performance consistently declines and response time increases with problem difficulty, reflecting LLMs' sensitivity to longer reasoning chains, more demanding modeling, and higher computational effort.

**Step-level vs. Answer-level Evaluation.** As shown in Table 1, final-answer and step-level evaluations diverge sharply with problem difficulty. Final-answer accuracy drops by over 40% from easy to medium tasks and falls below 10% on hard problems, indicating that models struggle to sustain reliability through long reasoning chains. In contrast, step-level evaluation shows that models still gain partial credit by correctly handling significant parts of the derivation. Even on hard problems, they apply key principles or derive valid intermediate equations before failing at later stages.

These results demonstrate that final-answer scoring alone severely underestimates reasoning ability, whereas step-level evaluation provides a more faithful measure of process competence under com-

Table 1: Step-level Accuracy and Final-Answer Accuracy across difficulty levels (Easy, Medium, Hard, and Avg.) for evaluated models.

| Model | Reasoning | Easy | | | Medium | | | Hard | | | Avg. | | |
|---|---|---|---|---|---|---|---|---|---|---|---|---|---|
| | | Final | Step | Time | Final | Step | Time | Final | Step | Time | Final | Step | Time |
| *Open-source Chat LLMs* | | | | | | | | | | | | | |
| Deepseek-chat | F | 31.42 | 48.4 | 45.22 | 21.12 | 39.84 | 67.87 | 15.6 | 33.99 | 84.02 | 23.4 | 41.36 | 64.04 |
| Qwen2.5-72B-Instruct-Turbo | F | 21.34 | 35.17 | 23.76 | 7.19 | 19.92 | 29.92 | 3.07 | 13.53 | 32.54 | 11.32 | 23.81 | 28.36 |
| Llama-3.3-70B-Instruct-Turbo | F | 18.38 | 31.84 | 8.61 | 8.99 | 19.73 | 10.79 | 4.35 | 16.47 | 11.63 | 11.18 | 23.35 | 10.21 |
| Llama-4-Scout-17B-16E | F | 19.72 | 30.54 | 57.22 | 8.16 | 15.83 | 58.36 | 4.13 | 10.33 | 59.99 | 11.35 | 19.78 | 58.40 |
| *Open-source Reasoning LLMs* | | | | | | | | | | | | | |
| Deepseek-Reasoner | T | 30.43 | 48.68 | 200.93 | 21.57 | 42.79 | 262.44 | 15.86 | 37.22 | 311.34 | 23.25 | 43.39 | 253.49 |
| GPT-OSS-20B | T | 15.61 | 25.2 | 10.21 | 6.52 | 14.46 | 12.17 | 2.3 | 6.79 | 12.63 | 8.72 | 16.27 | 11.57 |
| GPT-OSS-120B | T | 17.39 | 28.49 | 16.07 | 8.54 | 20.13 | 23.40 | 4.35 | 13.4 | 26.01 | 10.66 | 21.32 | 21.39 |
| Qwen3-235B-A22B-Instruct | T | 21.74 | 33.7 | 22.29 | 8.31 | 18.07 | 29.95 | 3.07 | 9.45 | 34.25 | 11.85 | 21.45 | 28.31 |
| *Proprietary Chat LLMs* | | | | | | | | | | | | | |
| Claude-sonnet-4 | F | 24.7 | 38.94 | 23.88 | 13.71 | 28.41 | 26.95 | 9.21 | 20.88 | 27.32 | 16.54 | 30.19 | 25.90 |
| GPT-4o-mini | F | 17.79 | 36.45 | 13.41 | 7.64 | 23.52 | 15.25 | 3.07 | 19.1 | 16.31 | 10.14 | 27.11 | 14.86 |
| GPT-4.1 | F | 24.51 | 40.8 | 18.14 | 11.69 | 23.36 | 28.95 | 4.35 | 14.67 | 35.56 | 14.39 | 27.4 | 26.80 |
| *Proprietary Reasoning LLMs* | | | | | | | | | | | | | |
| Gemini-2.5-Flash | T | 23.52 | 36.5 | 41.08 | 14.61 | 27.52 | 58.40 | 10.74 | 24.72 | 72.86 | 16.84 | 30.09 | 56.08 |
| Gemini-2.5-Pro | T | 32.41 | 49.4 | 60.33 | 19.1 | 37.63 | 77.37 | 18.67 | 34.61 | 88.82 | 23.99 | 41.19 | 74.28 |
| GPT-5 | Low | 33.0 | 54.76 | 26.77 | 22.92 | 44.67 | 33.86 | 15.86 | 36.76 | 40.25 | 24.66 | 46.17 | 33.05 |
| GPT-5 | Medium | 30.83 | 48.03 | 50.12 | 17.3 | 34.92 | 66.96 | 10.49 | 26.97 | 83.67 | 20.42 | 37.55 | 65.48 |
| GPT-5 | High | 37.75 | 58.36 | 103.59 | 26.29 | 52.7 | 127.59 | 21.99 | 50.28 | 153.32 | 29.36 | 54.13 | 126.04 |
| GPT-5-mini | Low | 25.3 | 43.15 | 20.78 | 17.98 | 38.1 | 27.10 | 11.0 | 27.85 | 30.30 | 18.71 | 37.02 | 25.65 |
| GPT-5-mini | Medium | 30.43 | 49.51 | 45.41 | 16.63 | 36.46 | 60.85 | 9.46 | 26.45 | 73.56 | 19.74 | 38.46 | 58.73 |
| GPT-5-mini | High | 34.78 | 56.82 | 149.47 | 23.82 | 48.64 | 178.56 | 17.14 | 38.53 | 210.08 | 26.01 | 48.78 | 176.78 |
| Grok-4 | T | 30.63 | 52.48 | 181.66 | 22.47 | 46.91 | 235.58 | 14.87 | 41.0 | 281.62 | 23.34 | 47.29 | 228.63 |
| o4-mini | T | 25.3 | 38.77 | 26.73 | 14.16 | 29.32 | 33.20 | 8.44 | 23.08 | 36.50 | 16.69 | 31.07 | 31.72 |
| *Multimodal Large Language Models* | | | | | | | | | | | | | |
| Claude-sonnet-4 | F | 27.72 | 47.32 | 19.46 | 16.06 | 37.02 | 22.77 | 11.63 | 32.85 | 23.37 | 19.19 | 39.71 | 21.69 |
| Gemini-2.5-Flash | T | 35.18 | 55.96 | 45.19 | 24.27 | 48.41 | 65.65 | 19.95 | 39.7 | 81.96 | 27.12 | 48.72 | 62.69 |
| Gemini-2.5-Pro | T | 36.96 | 57.66 | 63.03 | 25.39 | 48.88 | 80.71 | 21.74 | 43.03 | 93.58 | 28.69 | 50.49 | 77.79 |
| GPT-4.1 | F | 32.02 | 55.41 | 23.10 | 20.9 | 44.55 | 33.15 | 11.25 | 36.36 | 40.82 | 22.28 | 46.26 | 31.60 |
| GPT-5 | Medium | 37.45 | 60.24 | 63.29 | 26.02 | 52.46 | 87.70 | 21.56 | 49.12 | 104.99 | 29.05 | 54.43 | 83.49 |
| GPT-5-mini | Medium | 29.9 | 48.06 | 48.88 | 19.91 | 41.6 | 64.30 | 13.95 | 34.4 | 75.42 | 21.96 | 41.96 | 61.69 |
| Grok-4 | T | 31.55 | 52.73 | 180.19 | 21.8 | 47.73 | 231.76 | 15.13 | 43.64 | 316.77 | 23.53 | 48.42 | 237.11 |

plex tasks. Moreover, step-level signals open promising avenues for training and data curation: If evaluation relies solely on final answers, rewards on difficult problems become extremely sparse. Instead, step-level scoring provides **rich intermediate reward signals**, offering valuable guidance for reinforcement learning and a principled basis for constructing higher-quality training data.

**Physics Domain Category Analysis.** We analyze LLM performance across physics domains and difficulty levels, as shown in Figure 4. Models exhibit varying accuracy across different types, with the highest performance observed in Thermodynamics and Statistical Physics and the lowest in Quantum Mechanics. Step-level evaluation further exposes weaknesses in reasoning coherence, and accuracy consistently drops from Easy to Hard problems across all domains.

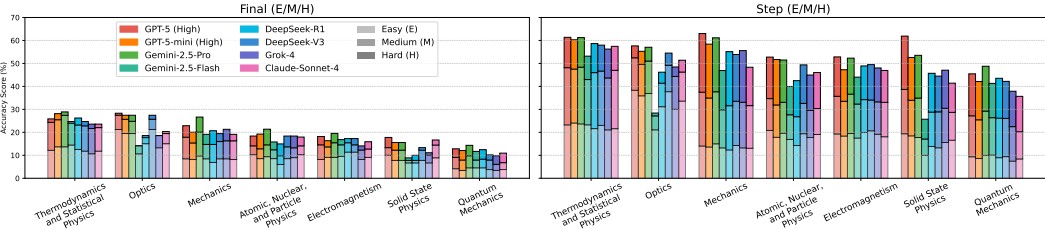

Figure 4: Step-level and final-answer accuracy across Physics Domain Categories and Difficulty Levels.

## 6.3 MODALITY AND REASONING-LEVEL COMPARISONS

**Text Models vs. Multimodal Models.** The effect of multimodal input varies across model families. In general, adding images provides stronger gains at the step level than at the final-answer level, highlighting its role in supporting intermediate reasoning. However, for smaller or weaker models,

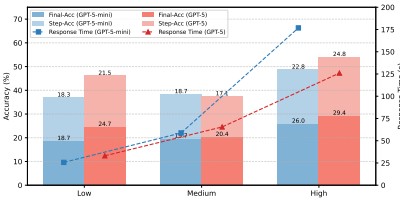

Figure 5: Comparison of accuracy and response time across reasoning levels.

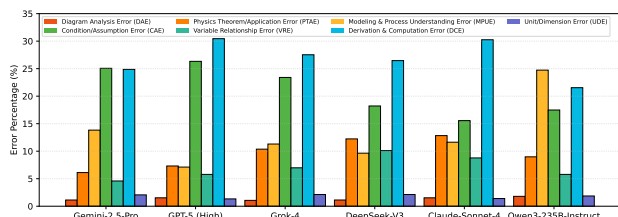

Figure 6: Distribution of primary error types across models

multimodal input can even be detrimental, as diagrams in physics problems often serve a presentational rather than informational role, with the critical content already conveyed in text. Figure is provided in Appendix E.2.

**Across Different Reasoning Level.** We observe that reasoning-oriented models exhibit consistently higher accuracy than chat-oriented models, but this improvement comes with substantially longer response times. We further evaluate GPT-5 and GPT-5-mini for three reasoning modes (*low*, *medium*, *high*), as shown in Figure 5. In general, results indicate an improvement in accuracy with increasing reasoning effort, along with reasoning latency. For GPT-5-mini, we observe performance improvement in final-accuracy and step-accuracy when increasing reasoning budget; meanwhile, the average latency of *medium* mode is 129.0% higher than the *low* mode, and the *high* mode is 589.2% higher. GPT-5 shows similar trend, though the *medium* mode performance drops below the *low* mode: this might be caused by over-thinking that is not thorough enough, while the *high* mode shows higher and more consistent performance. Notably, while o4-mini was previously claimed to be a good reasoning model, its performance here is relatively poor; one possible explanation is that, as a distilled model, it suffers from limited generalization and thus struggles with complex reasoning tasks beyond its training distribution.

### 6.3.1 ERROR ANALYSIS

We perform error analysis on the first incorrect step detected in each solution as shown in Figure 6, using a unified taxonomy that integrates process-level physics reasoning errors with formula-level derivation errors. The classification covers seven categories (detailed definitions are provided in Appendix G.2): (1) Diagram Analysis Error (DAE), (2) Physics Theorem Application Error (PTAE), (3) Modeling and Process Understanding Error (MPUE), (4) Condition or Assumption Error (CAE), (5) Variable Relationship Error (VRE), (6) Derivation and Computation Error (DCE), and (7) Unit Dimension Error (UDE).

The dominant error types across models are Condition/Assumption Errors (CAE), which arise when models set up inconsistent or incorrect physical assumptions; Derivation & Computation Errors (DCE), which occur when models make mistakes in algebraic manipulation or calculation; and Modeling & Process Understanding Errors (MPUE), which reflect failures in mapping the problem into the correct physical model or reasoning process. This indicates that LLMs often fail both in establishing consistent physical conditions and in executing algebraic reasoning.

### 6.4 EVALUATION FRAMEWORK ANALYSIS

We further evaluated PRISM-DAG with human annotations to assess effectiveness.

**Annotation Setup.** We randomly sampled 70 problems (10 from each domain) along with their corresponding DeepSeek-V3 (text-only) solutions. Each problem–solution pair was independently evaluated by two human experts to reduce variance , with annotators including an IPhO Gold Medalist and Top-Tier Physics PhD. In cases where the two experts' scores differed substantially, a third annotator was invited to adjudicate and determine the final score.

**Results.** We quantified the agreement between framework-generated scores and human annotations using Kendall's $\tau_b$ correlation coefficient, along with statistical significance testing via both asymptotic and permutation-based p-values (see Appendix F.2 for details). Higher $\tau_b$ values indicate stronger concordance, with significance levels verifying the robustness of the observed correlations.

Table 2 demonstrates the clear superiority of PRISM-DAG. which achieves the highest $\tau_b$ and lowest $p$-values. *LLM-as-Judge* is purely outcome-based, assigning only binary $0/1$ scores, while *PSAS-S*, though process-based, evaluates steps independently without modeling causal dependencies. Both baselines are LLM-based, whereas our non-LLM PRISM-DAG explicitly accounts for causality across steps, leading to stronger alignment with human judgments. We analyzed failure cases from our evaluator and two baselines to understand strengths and limitations. Details are provided in Appendix F.3.

Table 2: Comparison of annotation alignment.

| Method | $\tau_b \uparrow$ | Asymptotic $p$-value $\downarrow$ | Permutation $p$-value $\downarrow$ |
|---|---|---|---|
| LLM-as-Judge | 0.294 | $6.90 \times 10^{-3}$ | $6.00 \times 10^{-3}$ |
| PSAS-S | 0.213 | $2.20 \times 10^{-2}$ | $2.09 \times 10^{-2}$ |
| **PRISM-DAG** | **0.346** | $\mathbf{1.31 \times 10^{-4}}$ | $\mathbf{1.00 \times 10^{-4}}$ |

## 7    CONCLUSION AND FUTURE WORK

We introduced PRISM-PHYSICS, a benchmark and a process-level evaluation framework that encodes physics solutions as DAGs and employs rule-based symbolic equivalence checking for reliable, fine-grained scoring.

Experiments reveal persistent reasoning failures in frontier LLMs, underscoring the challenge of sustaining coherent derivations in physics. PRISM-PHYSICS establishes a principled and interpretable foundation for process-level evaluation, enabling more robust benchmarks and advancing LLMs toward deeper scientific reasoning, while its step-level signals provide both training guidance and a principled basis for higher-quality data.

Although PRISM-PHYSICS currently focuses on physics, our evaluation framework is domain-agnostic and can be readily extended to other subjects such as mathematics, chemistry, and biology. In future work, we also plan to use the benchmark for post-training LLMs, particularly to study the benefits of incorporating process-level signals during RL-based fine-tuning.

Furthermore, our framework is designed to be easily adapted to existing datasets. We therefore encourage both current and future benchmark developers to adopt our framework alongside their original evaluation methods, in order to provide more comprehensive and consistent assessments.

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

# Supplementary Materials for PRISM-Physics: Causal DAG-Based Process Evaluation for Physics Reasoning

## A  DAG STRUCTURE DETAILS

### A.1  PROOF OF THEOREM 1

*Proof.* (i) If $\Phi(\vdash_1) = \Phi(\vdash_2)$, then their edge sets coincide: $E_{\vdash_1} = E_{\vdash_2}$. By definition of $E_\vdash$, this is equivalent to

$$\{(A, B) : A \vdash_1 B\} = \{(A, B) : A \vdash_2 B\},$$

hence $\vdash_1 = \vdash_2$.

(ii) If $\Psi(G_1) = \Psi(G_2)$, then their kernels coincide: $\vdash_{E_1} = \vdash_{E_2}$. Unwinding the definition,

$$(A, B) \in E_1 \iff A \vdash_{E_1} B \iff A \vdash_{E_2} B \iff (A, B) \in E_2,$$

so $E_1 = E_2$ and thus $G_1 = G_2$.

Order-keeping ensures that $\Phi(\vdash) \in \mathsf{DAG}$ (acyclic by Assumption 2; all edges point forward), and that $\Psi(G) \in \mathsf{Just}$ (singletons and forward edges by construction). Finally,

$$\Psi(\Phi(\vdash)) = \vdash \quad \text{and} \quad \Phi(\Psi(G)) = G$$

hold by definition of $E_\vdash$ and $\vdash_E$.  □

### A.2  PROOF OF THEOREM 2

*Proof.* ($\subseteq$). Soundness gives

$$\mathsf{S}(\mathcal{M}) \subseteq \mathrm{Ach}(\mathcal{M}).$$

($\supseteq$). Matched Inclusion gives

$$\mathcal{M} \subseteq \mathsf{S}(\mathcal{M}).$$

By Ancestor Closure, if $B \in \mathsf{S}(\mathcal{M})$ and $A \prec B$, then $A \in \mathsf{S}(\mathcal{M})$. Applying this repeatedly from every $B \in \mathcal{M}$ along all reverse paths implies

$$\mathrm{Anc}(\mathcal{M}) \subseteq \mathsf{S}(\mathcal{M}).$$

Hence

$$\mathrm{Ach}(\mathcal{M}) \subseteq \mathsf{S}(\mathcal{M}).$$

Combining both directions yields the equality:

$$\mathsf{S}(\mathcal{M}) = \mathrm{Ach}(\mathcal{M}).$$

□

## B  EVALUATION FRAMEWORK DETAILS

### B.1  EQUIVALENCE MATCHING DETAILS

Given a standard solution (composed of multiple formulas in a DAG structure) and an LLM solution (composed of multiple formulas), we compare every possible pair of one solution-formula and one LLM-formula. In practice, this process, which runs on CPU cores, can be efficiently parallelized, hence its time consumption is very small compared to other steps (e.g. serialized generation of tokens with LLMs) in our benchmarking pipeline. The rest of this section discusses in detail how we compare two physics formulas.

As described, checking whether two physics formulas are equivalent faces several critical difficulties, where we provide a more detailed discussion:

1. **Equivalence of equations.** Formula Matching (i.e. checking equivalence of two formulas) is harder than Expression Matching (i.e. checking equivalence of two expressions). Previous work for expression matching mainly use tree-based formula parsing (Qiu et al., 2025). However, such tree-based parsing is not powerful enough to compare formulas.

   For result-based judges or expression-based judges, one could check equivalent expressions only, but checking equivalent formulas is critical for more fine-grained process-based score.

2. **Constant substitutions.** In physics two equivalent variables might be written in different forms. For example, the coulomb force $F = kQq/r^2$ can be written as $F = Qq/(4\pi\epsilon_0 r^2)$: these two expressions are equivalent, but they are in different forms. Sometimes universe constants can be expressed in detailed numbers, for example: $E = mc^2 = m * (3.0 * 10^8 m/s)^2$

3. **Unit conversion.** Values can be expressed in different units. For example, $f = 50$ Hz $= 50$ s$^{-1}$, using different units results in equivalent but seemingly different formulas.

We show our proposed algorithm for formula equivalence matching (i.e., comparing whether two equations are equivalent) in Algorithm B.2.

First, we conduct substitution of certain variables: this includes Math constants (e.g. $\pi, e$,etc.), Physics constants (e.g. $k = \frac{1}{4\pi\varepsilon_0}, c_0 = \frac{1}{\sqrt{\varepsilon_0 \mu_0}}$,etc.), constants or values provided in the problem (e.g. provided length, etc.), described as 'Stage 1' in the algorithm.

After unifying all variables across formulas, we move forward to the iterated process of choosing a target variable – randomly assign values to other variables – solving for the target variable. The equivalence of solution sets are used as proxy for comparing equivalence of these two equations. In practice, we conduct at most $N_{max} = 40$ iterations. In each iteration, one target variable is selected randomly, and other variables are assigned random values in $[2, 20]$. Each iteration would generate one out of three possible outcomes: (1) not-rejecting equivalence trial; (2) rejecting equivalence trial; and (3) failure trial (in cases when both equations have no solutions). For each iteration, if both equations have solutions and their solution sets are equal within some tolerance threshold (relevant difference $\varepsilon = 10^{-6}$), it is classified as (1) not-rejecting equivalence trial; if their solution number is different or their solution sets are not equal within certain threshold, this trial is classified as (2) rejecting equivalence trial; while if both equations provide no solutions, such trial is classified as (3) failure trial. We continue to run iterations until we have enough successful solutions (at least $N_{succ} = 10$) or we reach the maximum number of iterations ($N_{max} = 40$). Then we reject the equivalence if the number of successful trials is less than $N_{succ} = 10$ or there exists a trial that is a rejection equivalence trial. [1].

Here we discuss some of these design choices.

**More-than-one Iterations.** In common cases, given sufficient tolerance, the $p - value$ of one iteration is small enough to reject the equivalence of non-equivalent equations. In practice, the tolerance is set to $10^{-6}$ considering possible computing errors. However, this would lead to false non-rejecting in special cases, when using only one iteration. Consider these two formulas: $f1 : x = A_0 + A_1 t^2 \delta$ and $f2 : x = A_0 + 2A_1 t^2 \delta$ where $A_0 \sim A_1 \sim 1, t \sim 1, \delta \sim 10^{-8}$. If we select $x$ or $A_0$ as target variable and randomly assign values to other variables, the difference in solutions of these two formulas would lie within the tolerance, and hence it cannot reject nonequivalence of these two formulas. In this case, using $A_1$ or $t$ as the target variable would reject the equivalence, making false non-rejecting rate of one iteration be around 0.5, which is way larger than what we expect. In our multi-iteration settings, the false non-rejecting rate of the 10-iteration examine process would be around $10^{-3}$ in this case, which is satisfactory enough.

**Positive Sampling Intervals.** Here we randomly sample most variables in a positive interval, i.e., $[2, 20]$. This is because most variables in physics problems are provided as positive ones, and using negative intervals for sampling may lead to false rejections. For example, $T = 2\pi\sqrt{a^3/GM}$ and $T = 2\pi a \sqrt{a/GM}$ (Kepler's Third Law) are considered equivalent in most physics settings: the semi-major axis $a$ of planet orbits should always be positive. Here, we simply set our sampling interval to be positive to avoid false rejections caused by similar reasons.

---

[1]Some optimizations for reducing time consumption are also used in practice (e.g. rejecting equivalence of formulas once after the first rejecting equivalence trial), which are trivial and have no impact to the output of this algorithm, hence we omit them here for clarity

## B.2 ALGORITHM OF EQUIVALENCE MATCHING

---

**Algorithm 1:** Equivalence Check for Two Physics Equations

---

**Input** : Two equations $E_1, E_2$ (symbolic); constants map $\mathcal{C}$ (symbol $\mapsto$ symbol or value); sampling range $R = [a, b]$; tolerance $\varepsilon$; limits $T_{\max}$ (time), $N_{\max}$ (trials), $N_{\text{succ}}$ (min successful trials), $N_{\text{eq}}$ (min equalities).

**Output:** Equivalent $\in$ EQUIVALENT, INEQUIVALENT; diagnostics $(n_{\text{eq}}, n_{\text{neq}}, n_{\text{fail}})$.

**Stage 1: Constant Variable Substitution**; **Procedure** SUBSTITUTECONSTANTS($E, \mathcal{C}$)
    Substitute all string-valued entries in $\mathcal{C}$ into $E$ ;      `// units/expressions; one pass, no recursion`
    Substitute all numeric-valued entries in $\mathcal{C}$ into $E$ ;      `// e.g., e, c₀, x_f`
    **return** updated $E$
$E_1 \leftarrow$ SUBSTITUTECONSTANTS($E_1, \mathcal{C}$); $E_2 \leftarrow$ SUBSTITUTECONSTANTS($E_2, \mathcal{C}$);

**Stage 2: Equivalent Check by Solving in Random Conditions**;
  $n_{\text{eq}} \leftarrow 0,; n_{\text{neq}} \leftarrow 0,; n_{\text{fail}} \leftarrow 0,; k \leftarrow 0; V \leftarrow$ free variables appearing in $E_1 \cup E_2$;
  **Function** SOLVETARGET($E, x^\star, \theta, T_{\max}$)
    **return** solution set $\mathcal{S}$ of $E$ for target $x^\star$ under assignment $\theta$ to $V \setminus x^\star$ within time $T_{\max}$
  **Function** ALLCLOSE($\mathcal{S}_1, \mathcal{S}_2, \varepsilon$)
    **return** TRUE iff the two finite sets match pairwise within relative/absolute tolerance $\varepsilon$
  **while** $k < N_{\max}$ **do**
    $k \leftarrow k + 1$; pick $x^\star \sim \text{Unif}(V)$ ;     `// random target variable`
    sample $\theta(v) \sim \text{Unif}(R)$ for each $v \in V \setminus x^\star$ ;     `// random values for non-target variable`
    $\mathcal{S}_1 \leftarrow$ SOLVETARGET($E_1, x^\star, \theta, T_{\max}$) ;
    $\mathcal{S}_2 \leftarrow$ SOLVETARGET($E_2, x^\star, \theta, T_{\max}$) ;
    **if** $\mathcal{S}_1$ or $\mathcal{S}_2$ is nonempty **then**
      **if** ALLCLOSE($\mathcal{S}1, \mathcal{S}2, \varepsilon$) **then**
        $n_{\text{eq}} \leftarrow n_{\text{eq}} + 1$ ; `// current trial does not reject equivalence`
      **else**
        $n_{\text{neq}} \leftarrow n_{\text{neq}} + 1$ ;     `// current trial rejects equivalence`
      **end**
    **else**
      $n_{\text{fail}} \leftarrow n_{\text{fail}} + 1$ ; `// current trial fails and cannot be used for judging equivalence`
    **end**
    **if** $(n_{\text{eq}} + n_{\text{neq}}) \geq N_{\text{succ}}$ **and** $k \geq N_{\text{succ}}$ **then**
      **break**
    **end**
  **end**

**Decision**; **if** $n_{\text{eq}} \geq N_{\text{eq}}$ **and** $n_{\text{neq}} = 0$ **then**
  **return** EQUIVALENT ;   `// Enough valid trials and no trial rejects equivalence`
**end**
**else**
  **return** INEQUIVALENT ; `// Exists trial rejecting equivalence or no enough valid trials`
**end**

---

## B.3 Algorithm of Scoring Pipeline

---

**Algorithm 2:** PRISM-DAG: Evaluation via DAG-Structured Rubric (3 Steps)

---

**Input** : Reference DAG $G = (V, E)$; each node $v \in V$ has formula $\phi(v)$ and prerequisite set $\mathrm{Pred}(v)$; student solution text $S$.

**Output:** Score $s \in [0, 1]$; matched set $M \subseteq V$; achieved set $A \subseteq V$.

**Step 1: Extract and Normalize Student Formulas**;

$\widehat{F} \leftarrow \mathrm{ExtractFormulas}(S)$ ;                              `// raw math expressions`

$F \leftarrow \emptyset$;

**foreach** $f \in \widehat{F}$ **do**

    $g \leftarrow \mathrm{Canonicalize}(f)$ ;

    **if** $\mathrm{IsValid}(g)$ **then**

        | $F \leftarrow F \cup \{g\}$

    **end**

**end**

**Step 2: Match to Reference DAG (Reference Nodes Only)**;

$M \leftarrow \emptyset$;

**foreach** $v \in V$ **do**

    **if** $\exists\, g \in F$ *s.t.* $\mathrm{Equivalent}(\phi(v), g)$ **then**

        | $M \leftarrow M \cup \{v\}$

    **end**

**end**

**Step 3: Dependency Tracing and Scoring**;

$A \leftarrow \emptyset$;

**Procedure** MARKANCESTORS($u$)

    **if** $u \notin A$ **then**

        $A \leftarrow A \cup \{u\}$;

        **foreach** $p \in \mathrm{Pred}(u)$ **do**

            | MARKANCESTORS($p$)

        **end**

    **end**

**foreach** $v \in M$ **do**

    | MARKANCESTORS($v$)

**end**

$s \leftarrow |A| \,/\, |V|$

**return** $(s, M, A)$;

---

## B.4 Scoring Example

Figure 7 illustrates an example of how a student's (or LLM's) solution is scored using formula matching and DAG-based back-propagation scoring.

**Step A (Formula Matching):** each formula in the student's solution is aligned one-to-one with the reference solution, with equivalent formulas connected by green lines. For clarity, not all formula pairs are drawn. Grey arrows denote the dependency relations (DAG) between formulas in the reference solution.

**Step B (Back-Propagation scoring):** once a derived formula in the DAG is matched, correctness is propagated backwards along the dependency graph, allowing upstream formulas to be credited as well. Correctly credited formulas are highlighted in orange, and the back-propagation path is indicated by orange arrows.

**Step C (Score Calculation):** the final score is calculated by adding the points of correctly credited formulas. In this example, the student achieves a score of $90/100$.

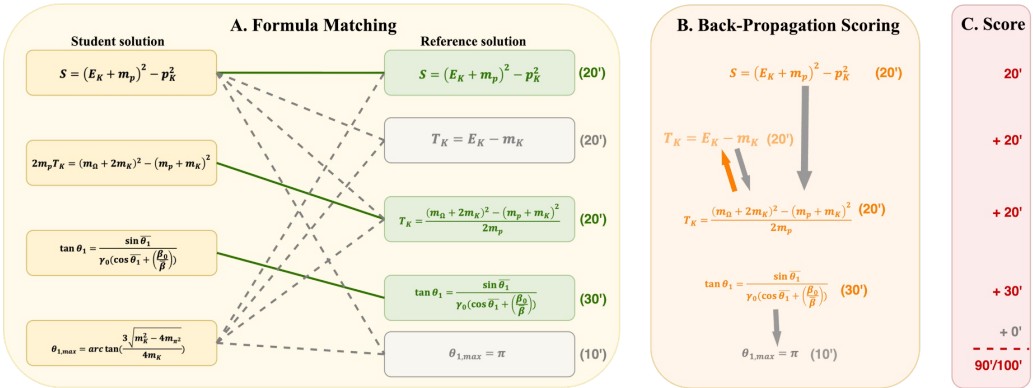

Figure 7: Scoring pipeline example. A) Formula matching aligns student and reference formulas. B) Back-propagation grading highlights correctly credited formulas along the dependency DAG. C) The final score is computed as the sum of credited points, yielding $90/100$ in this case.

## C  DATASET CURATION DETAILS

### C.1  DATA SOURCE AND COLLECTION

Our primary source of questions and detailed step-by-step solutions is the book *Major American Universities Ph.D. Qualifying Questions and Solutions*. The problems were extracted from PDF format, reorganized in Markdown for readability, and further converted into JSON for structured storage. Notably, this book has been widely adopted as a training resource for advanced physics competitions, ensuring both the difficulty and the pedagogical value of the collected problems. Problems requiring purely textual answers (e.g., "Describe . . .", "Is . . . stable?") were set aside in a separate collection and are excluded from the current framework.

### C.2  THREE-STEP REWRITING PIPELINE DETAILS

To ensure consistency and evaluability, each sample is processed through a three-stage pipeline:

1. **Formula normalization.** All mathematical expressions are rewritten into a uniform canonical format in LaTeX. This normalization supports precise symbolic equivalence checking, preventing mismatches due to notational variation. For numerical problems, explicit rules on the number of significant figures are enforced, ensuring consistent standards across all answers.

2. **Context clarification.** Each problem statement is rewritten to make all variable definitions and final answer requirements explicit. Where the original text leaves meanings implicit (e.g., undefined symbols or missing constants), clarifications are added to resolve ambiguities. The result is a self-contained problem statement that can be understood without external assumptions.

3. **DAG construction.** Each worked-out solution is converted into a *directed acyclic graph* (DAG) of formulas according to the requirement in Section 3.3. We first verify if it satisfies the requirements via rule-based methods before the LLM-based verification.

**Verification and Quality Control**   At the end of each stage, we employ an LLM-based verification module to check compliance with formatting, clarity, and dependency rules. If the verification fails—such as when ambiguities persist in variable definitions or the DAG contains extraneous steps—the stage is repeated with targeted corrective feedback. This iterative loop of generation and verification ensures robustness, yielding uniformly high-quality results across the dataset.

**Fine-Grained Enhancements**   Beyond the main pipeline, several additional refinements were systematically applied:

- **Numerical precision.** For problems with numerical answers, explicit enforcement of significant-figure rules was introduced in the problem statement.
- **Explicit constants.** All physical constants and context-dependent variables appearing in either the problem or the solution were explicitly defined in the rewritten version.
- **Answer formatting.** Uniform formatting standards for final answers were applied to the problem context, including required units and symbolic representations.

Individually, these refinements may appear minor; collectively, they substantially improve the machine-actionability, reliability, and pedagogical clarity of the dataset.

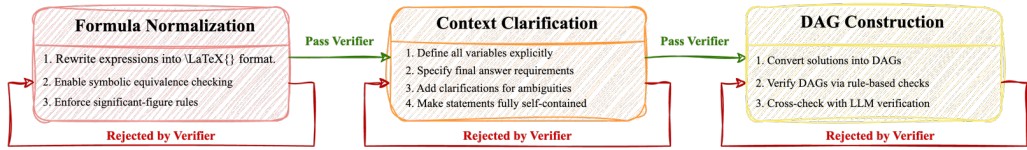

Figure 8: Overview of the Three-Step Rewriting Pipeline

## C.3 Prompts for Three-Step Rewriting Pipeline

Here are all the prompts we adopt for rewriting.

---

**Format Instructions**

1) **One Formula Per Block**
  - Each formula must be wrapped in its own `$$...$$` block.
  - Avoid chaining multiple equalities or expressions in a single block.
  - Exception: Chained variable comparisons in inequalities are allowed **only if explicitly required**.
2) **No Terminal Punctuation**
  - Do not end any formula block with punctuation marks (e.g. `,`, `.`, `;`).
3) **SI Unit Format**
  - Always write units using `\unit{}` to ensure proper parsing (e.g., `\unit{m}`, `\unit{m/s}`). Notice that the numbers should be put outside the unit, i.e. use `$3\unit{km/h}$` instead of `$\unit{3km/s}$`.
4) **Strip Extra Formatting Commands**
  REMOVE the following:
  - Delimiters: `\left`, `\right`
  - Fonts/Styles: `\mathrm`, `\mathit`, `\mathbf`, `\text`
  - Spacing: `\,`, `\;`, `\!`, `~`, `\quad`, `\qquad`
  - Multi-line: `\begin{aligned}...\end{aligned}`
5) **Standard Calculus Notation**
  Use canonical forms for all calculus expressions:
  - Derivatives: `\frac{dy}{dx}`
  - Partial derivatives: `\frac{\partial f}{\partial x}`
  - Integrals: `\int_0^t v dt` (omit spacing commands)
  - Summations: `\sum_{i=1}^n x_i`

---

**Rewriting Prompt - First Stage**

You are an expert in Physics, and you are going to rewrite a given problem and solution into a standard form. The formatting requirements are below:

**Format Instructions**
Moreover, you should make sure that the answers can be graded correctly, you should make sure the written form of the final answer is unified, which means: 1) You should make sure all variables needed in the solution, no matter in the final answer or in the intermediate steps, are defined or specified in the problem, either in the context or in the subproblems. e.g. 'You should use $E_k$ for kinetic energy, $E_p$ for potential energy, $E$ for total energy, $M$ for the mass of the central body, $m$ for the mass of the satellite, $R$ for the radius of the orbit.'

---

2) You should make sure all variables and concepts are defined clearly in the problem.

3) For all accurate values, don't write them in decimals but fractions instead. For example instead of $y = 2.25x$ you should write $y = \frac{9}{4}x$. However if you believe the value is approximate (for example you think 2.25 is not an accurate value) then you should leave it as decimals.

4) You should try to avoid putting a representation in a formula block, but instead use equations or inequalities. For example instead of writing "The maximum energy is $2E_0$", you should write $E_{max} = 2E_0$ and put the definition of $E_{max}$ in the problem context. Even if it is hard to represent it with a single variable, you should write $ans = \cdots$, and mention that the final answer should be written in this form in the problem context.

Now according to the requirements, please rewrite the problem and solution below.

**The sample**

Your output should be in the same json format, keeping all entries even if unmodified. Don't forget any single entry or your output would be invalid. Your response should be formatted as ```json\n<your_rewritten_json>\n```, and nothing other than the rewritten json should be in your output.

---

## Rewriting Prompt - Second Stage

You are an expert in Physics, and you are going to rewrite a given problem to make it clearer. More specifically, we wish to clarify the requirement for the final answer of each problem, which means:

1) You should add an entry 'final_answer_form' which has three options 'algebraic', 'numerical' and 'text-based' representing the form of the final answer. 'text-based' means the final answer is not a calculation result or a derived formula, but instead a text description or statement. Notice that this only depends on the final answer in the standard solution, therefore each subquestion can only have one final_answer_form. If this is the same for all subproblems, you may write it as an entry of the whole problem, else you may add an entry for each subproblem separately. In any way, this should be a separate entry as item['final_answer_form'] for the whole problem or item['subquestions'][i]['final_answer_form'] for the i-th subquestion.

A hint on how to decide if an answer is algebraic or numerical: if the answer is a formula with multiple variables and each side of the formula has variables, it is algebraic; if there is only one variable or variables only exist in one side, it is numerical (e.g. $v_s=3\times 10^7\unit{m/s}$, $\rho_{min}\approx 1.5\times 10^3 \unit{kg/m^3}$ are both numerical answers.) Only the final formula would decide if it is algebraic or numerical.

2) You should add an entry 'final_answer_instructions' based on the final answer form. Similar to 'final_answer_form', it should be either an entry of the whole problem (if the instruction is suitable for the whole problem) or separate entries for each subquestion. The instruction should contain the following information based on the final answer form:

a) If the final answer is algebraic, you should specify the format of the final answer in the problem, e.g., 'Your final answer should be given in the form of $v_{min} = ...$, and your response should be written with $H, T, m, g$'. The variable requirement for the final answer should fully match the final answer in the standard solution and make there's no redundant variables (for example, if $E_k = \frac{1}{2}mv^2$, then you should at most provide two variables among $E_k, m, v$ for the final answer)

b) If the final answer is numerical, you should instead write 'Your final answer should express ... as a numerical value', and if the final answer is an approximate value, you should also specify the number of significant figures needed according to the standard answer. Moreover you should add an entry like 'significant_figures': 3 to the problem or subproblem.

c) If the final answer is text-based, you should try to restrict the form of final statement, for example you should write 'Your final answer should be 'The equilibrium is stable/unstable.'', so that we can seek for the sentence to judge the correctness of the student answer. Now according to the requirements, please rewrite the problem and solution below.

**The sample**

Your output should be in the same json format, keeping all entries even if unmodified, and add your new entries as required. Don't forget any single entry or your output would be invalid. Make sure you output a valid json. DO NOT put any hint for the final answer in the instruction! Only give some information to regularize the format!

Your response should be formatted as ```json\n<your\_rewritten\_json>\n```, and nothing other than the rewritten json should be in your output.

**Rewriting Prompt - Third Stage**

You are an expert in Physics, and you are setting up a grading standard for a given problem. More specifically, your job is to find the 'core formulas' in the solution. These core formulas should reflect some significant progress in solving the problem and thus you think they are worth some credit. Moreover, you have to organize them in a given format to set up a grading standard.

1) You should organize the core formulas in a list, each formula represented by a dict including the following entries:

```
{
"index": the index of the formula, counting from 1.
"formula": the content of the formula, which should be a string
    wrapped with double-dollar ($$) symbols.
"dependency": this is the crucial part of the grading standard. The
    dependency should be a list of indices showing which previous
    formulas this one depends on, which indicates without those
    previous formulas this formula can't be derived. Notice that only
    direct causalities should be considered, for example if A leads to
     B and B leads to C, you don't need to put A in the dependency of
    C. A formula can never depend on another formula after it.
"is_final_answer": (optional) this is true if this formula is the
    final answer of a subproblem or the whole problem. The last
    formula among all should always be a final answer.
}
```

2) Every formula in the grading standard must exist in the original solution, you should not create any new formula in the grading standard, and you should not make any modification to the formulas: just directly copy them from the solution.

3) You should ensure that for an isolated question or subquestion, if the final answer is correct, the student should receive full score for it. That means, any formula should be 'reachable' from the final answer (or at least the final answer of one subproblem) in the dependency graph. If the final answer contains more than one formuls, you may simply mark multiple formulas as final answers.

Below are some examples for you to better understand the requirement.

...

Now, according the requirement, please write the grading standard for the problem and solution below.

**The sample**

---

**Reviewing Prompt - First Stage**

You are a professor, and now you should review whether your assistant correctly rewrote the problem and solution into a standard form. More specifically, he should make sure the written form of the final answer is unified, so that the student answers can be graded correctly. Below was your instruction to him which he should satisfy:

**Problem Requirement as in the rewriting prompt**

Moreover, the formulas should also satisfy the following formatting requirements:

**Format Instructions**

You should return something like '`<judge>valid</judge><reason>...</reason>`' or '`<judge>invalid</judge><reason>...</reason>`'. You may think before your final output. Keep the reason concise.

Below are some examples for you to understand it better:

...

Now, according to the instructions, you should decide if your assistant has rewritten the problem and solution correctly. Below is the problem and solution rewritten by your assistant:

**The Sample**

---

The reviewing prompt for the second and third stage are basically the same with the first stage, only changing the requirement according to that in the rewriting prompt.

## C.4 REWRITING EXAMPLES

---

**PRISM-PHYSICS Rewriting Example - Stage 1**

**Problem**: A bucket of mass $M$ (when empty) initially at rest and containing a mass of water is being pulled up a well by a rope exerting a steady force $P$ The water is leaking out of the bucket at a steady rate such that the bucket is empty after a time $\tau$ .Find the velocity of the bucket at the instant it becomes empty.

**Original OCR Solution**:
Let the total mass of the bucket and water be $M'$ . Then

$$M' = M + m - \frac{mt}{T},$$

where $m$ is the initial mass of the water. As the leaking water has zero velocity relative to the bucket,the equation of motion is

$$M'\frac{dv}{dt} = P - M'g,$$

or

$$dv = \frac{P - M'g}{M'}dt = \left(\frac{P}{M + m - \frac{m}{T}t} - g\right)dt.$$

The velocity of the bucket at the instant it becomes empty is

$$v = \int_0^{} \frac{Pdt}{M + m - \frac{m}{T}t} - gT = \frac{PT}{m}\ln\left(\frac{M + m}{M}\right) \cdot gT .$$

**Error:**
1. The problem statement uses $\tau$, but the recognized solution incorrectly shows $T$
2. In the final equation, the minus sign $-$ was misrecognized as a multiplication symbol $\cdot$.

**Rewritten Solution** :
Let the total mass of the bucket and water at time $t$ be $M'$.

$$M' = M + m - \frac{mt}{\tau}$$

The equation of motion for the bucket (plus remaining water) is

$$M'\frac{dv}{dt} = P - M'g$$

Rearranging gives

$$dv = \left(\frac{P}{M + m - \frac{m}{\tau}t} - g\right)dt$$

The velocity $v$ of the bucket at time $t = \tau$ (when empty) starting from rest is

$$v_{empty} = \int_0^{\tau} \frac{Pdt}{M + m - \frac{m}{\tau}t} - g\tau$$

Evaluating the integral yields

$$v_{empty} = \frac{P\tau}{m}\ln\left(\frac{M + m}{M}\right) - g\tau$$

where $v_{empty}$ is the velocity of the bucket at the instant it becomes empty.

---

**PRISM-PHYSICS Full Sample Example**

```
{
    "id": 1001,
    "context": "A static charge distribution produces a radial electric field given by\n$$\n\
mathbf{E} = A \frac{e^{−b r}}{r} \mathbf{e}_r\n$$\nwhere $A$ is a constant with
units \unit{V\cdot m}, $b$ is a constant with units \unit{m^{−1}}, and $r$ is the radial
distance from the origin in \unit{m}. You should use $\rho$ for the charge density in \
unit{C/m^3}, $\varepsilon_0$ for the vacuum permittivity in \unit{C^2/(N\,m^2)}, $\
delta(\mathbf{r})$ for the Dirac delta function, and $Q$ for the total charge in \unit{C}.
When asked for the total charge, provide your answer as\n$$\nQ=...\n$$\nwhere $Q$
denotes the total charge.",
    "source": "",
    "images": [
      {
        "caption": "Fig. 1.1",
        "location": "..."
      }
    ],
    "subquestions": [
      {
        "letter": "a",
        "subproblem": "Find the charge density $\rho$ (in \unit{C/m^3}) that produces the
given electric field, including both the regular part as a function of $r$ and any singular (
delta function) contributions at the origin. State your result for $\rho$ explicitly. You may
sketch its form with reference to Fig. 1.1.",
        "solution": "The charge density $\rho$ is given by Gauss's law in differential form:\
n$$\n\rho = \varepsilon_0 \nabla \cdot \mathbf{E}\n$$\nThe electric field is\n$$\n\
mathbf{E} = A \frac{e^{−b r}}{r} \mathbf{e}_r\n$$\nThe divergence in spherical
coordinates for a radial function $f(r) \mathbf{e}_r$ is\n$$\n\nabla \cdot (f(r) \mathbf{
e}_r) = \frac{1}{r^2} \frac{d}{dr} (r^2 f(r))\n$$\nFor $f(r) = A \frac{e^{−b r}}{r}$, we
 compute\n$$\nf(r) = A \frac{e^{−b r}}{r}\n$$\n$$\nr^2 f(r) = A r e^{−b r}\n$$\n$$\
n\frac{d}{dr}(A r e^{−b r}) = A \frac{d}{dr}(r e^{−b r})\n$$\n$$\n\frac{d}{dr}(r e
^{−b r}) = e^{−b r} − b r e^{−b r}\n$$\n$$\n\frac{d}{dr}(A r e^{−b r}) = A e^{−b r} −
A b r e^{−b r}\n$$\n$$\n\nabla \cdot \mathbf{E} = \frac{1}{r^2} (A e^{−b r} − A b r
e^{−b r})\n$$\n$$\n\nabla \cdot \mathbf{E} = \frac{A e^{−b r}}{r^2} − \frac{A b e
^{−b r}}{r}\n$$\nHowever, the Laplacian of $\frac{1}{r}$ in three dimensions also
gives a delta function:\n$$\n\nabla^2 \left( \frac{1}{r} \right) = −4 \pi \delta(\mathbf{
r})\n$$\nSo the divergence, including the singularity at the origin, is\n$$\n\nabla \cdot
\mathbf{E} = −A b \frac{e^{−b r}}{r^2} + 4 \pi A \delta(\mathbf{r})\n$$\nTherefore,
the charge density is\n$$\n\rho = −\varepsilon_0 A b \frac{e^{−b r}}{r^2} + 4 \pi \
varepsilon_0 A \delta(\mathbf{r})\n$$\nThe final answer should be written as\n$$\n\
rho = −\varepsilon_0 A b \frac{e^{−b r}}{r^2} + 4 \pi \varepsilon_0 A \delta(\mathbf{r
})\n$$",
        "final_answer_form": "algebraic",
        "final_answer_instructions": "Your final answer should be given in the form $\rho = ...
$, and your response should be written only with $A, b, r, \varepsilon_0, \delta(\mathbf{r
})$."
      },
      {
        "letter": "b",
        "subproblem": "What is the total charge $Q$ (in \unit{C}) present for the above
charge density? Express $Q$ using the variables defined. Write the final answer using the
form $Q=...$",
        "solution": "The total charge $Q$ is given by\n$$\nQ = \int_{\mathbb{R}^3} \rho
dV\n$$\nWith $\rho = −\varepsilon_0 A b \frac{e^{−b r}}{r^2} + 4 \pi \varepsilon_0 A
\delta(\mathbf{r})$, we have\n$$\nQ = \int_{\mathbb{R}^3} \Bigl[−\varepsilon_0 A b
```

\frac{e^{−b r}}{r^2} \Bigr] dV + \int_{\mathbb{R}^3} 4 \pi \varepsilon_0 A \delta(\mathbf{r}) dV\n$$\nThe first term becomes (working in spherical coordinates):\n$$\n\int_{\mathbb{R}^3} −\varepsilon_0 A b \frac{e^{−b r}}{r^2} dV = −\varepsilon_0 A b \int_{0}^{\infty} \int_{0}^{\pi} \int_{0}^{2\pi} \frac{e^{−b r}}{r^2} r^2 \sin\theta d\phi d\theta dr\n$$\n$$\n= −\varepsilon_0 A b \int_{0}^{\infty} e^{−b r} dr \int_{0}^{\pi} \sin\theta d\theta \int_{0}^{2\pi} d\phi\n$$\n$$\n\int_{0}^{\infty} e^{−b r} dr = \frac{1}{b}\n$$\n$$\n\int_{0}^{\pi} \sin\theta d\theta = 2\n$$\n$$\n\int_{0}^{2\pi} d\phi = 2\pi\n$$\n$$\n−\varepsilon_0 A b \cdot \frac{1}{b} \cdot 2 \cdot 2\pi = −4\pi \varepsilon_0 A\n$$\nThe second term is\n$$\n\int_{\mathbb{R}^3} 4 \pi \varepsilon_0 A \delta(\mathbf{r}) dV = 4 \pi \varepsilon_0 A\n$$\nTherefore,\n$$\nQ = −4\pi \varepsilon_0 A + 4\pi \varepsilon_0 A\n$$\n$$\nQ = 0\n$$",
      "final_answer_form": "numerical",
      "final_answer_instructions": "Your final answer should be $Q=0$."
    }
  ],
  "grading_standard": [
    {
      "index": 1,
      "formula": "$$\rho = \varepsilon_0 \nabla \cdot \mathbf{E}$$",
      "dependency": []
    },
    {
      "index": 2,
      "formula": "$$\mathbf{E} = A \frac{e^{−b r}}{r} \mathbf{e}_r$$",
      "dependency": []
    },
    {
      "index": 3,
      "formula": "$$\nabla \cdot (f(r) \mathbf{e}_r) = \frac{1}{r^2} \frac{d}{dr} (r^2 f(r))$$",
      "dependency": []
    },
    {
      "index": 4,
      "formula": "$$f(r) = A \frac{e^{−b r}}{r}$$",
      "dependency": [
        2
      ]
    },
    {
      "index": 5,
      "formula": "$$r^2 f(r) = A r e^{−b r}$$",
      "dependency": [
        4
      ]
    },
    {
      "index": 6,
      "formula": "$$\frac{d}{dr}(A r e^{−b r}) = A \frac{d}{dr}(r e^{−b r})$$",
      "dependency": [
        5
      ]
    },
    {
      "index": 7,
      "formula": "$$\frac{d}{dr}(r e^{−b r}) = e^{−b r} − b r e^{−b r}$$",
      "dependency": []

```
    },
    {
     "index": 8,
     "formula": "$$\frac{d}{dr}(A r e^{-b r}) = A e^{-b r} - A b r e^{-b r}$$",
     "dependency": [
       6,
       7
     ]
    },
    {
     "index": 9,
     "formula": "$$\nabla \cdot \mathbf{E} = \frac{1}{r^2} (A e^{-b r} - A b r e^{-b r
})$$",
     "dependency": [
       3,
       8
     ]
    },
    {
     "index": 10,
     "formula": "$$\nabla \cdot \mathbf{E} = \frac{A e^{-b r}}{r^2} - \frac{A b e^{-
b r}}{r}$$",
     "dependency": [
       9
     ]
    },
    {
     "index": 11,
     "formula": "$$\nabla^2 \left( \frac{1}{r} \right) = -4 \pi \delta(\mathbf{r})$$",
     "dependency": []
    },
    {
     "index": 12,
     "formula": "$$\nabla \cdot \mathbf{E} = -A b \frac{e^{-b r}}{r^2} + 4 \pi A \
delta(\mathbf{r})$$",
     "dependency": [
       10, 11
     ]
    },
    {
     "index": 13,
     "formula": "$$\rho = -\varepsilon_0 A b \frac{e^{-b r}}{r^2} + 4 \pi \varepsilon_0
 A \delta(\mathbf{r})$$",
     "dependency": [
       1,
       12
     ],
     "is_final_answer": true
    },
    {
     "index": 14,
     "formula": "$$Q = \int_{\mathbb{R}^3} \rho dV$$",
     "dependency": []
    },
    {
     "index": 15,
```

```
      "formula": "$$Q = \int_{\mathbb{R}^3} \Bigl[-\varepsilon_0 A b \frac{e^{-b r}}{
r^2} \Bigr] dV + \int_{\mathbb{R}^3} 4 \pi \varepsilon_0 A \delta(\mathbf{r}) dV$$",
      "dependency": [
        13,
        14
      ]
    },
    {
      "index": 16,
      "formula": "$$\int_{\mathbb{R}^3} -\varepsilon_0 A b \frac{e^{-b r}}{r^2} dV =
-\varepsilon_0 A b \int_{0}^{\infty} \int_{0}^{\pi} \int_{0}^{2\pi} \frac{e^{-b r}}{r
^2} r^2 \sin\theta d\phi d\theta dr$$",
      "dependency": [
        15
      ]
    },
    {
      "index": 17,
      "formula": "$$= -\varepsilon_0 A b \int_{0}^{\infty} e^{-b r} dr \int_{0}^{\pi} \
sin\theta d\theta \int_{0}^{2\pi} d\phi$$",
      "dependency": [
        16
      ]
    },
    {
      "index": 18,
      "formula": "$$\int_{0}^{\infty} e^{-b r} dr = \frac{1}{b}$$",
      "dependency": []
    },
    {
      "index": 19,
      "formula": "$$\int_{0}^{\pi} \sin\theta d\theta = 2$$",
      "dependency": []
    },
    {
      "index": 20,
      "formula": "$$\int_{0}^{2\pi} d\phi = 2\pi$$",
      "dependency": []
    },
    {
      "index": 21,
      "formula": "$$-\varepsilon_0 A b \cdot \frac{1}{b} \cdot 2 \cdot 2\pi = -4\pi \
varepsilon_0 A$$",
      "dependency": [
        17,
        18,
        19,
        20
      ]
    },
    {
      "index": 22,
      "formula": "$$\int_{\mathbb{R}^3} 4 \pi \varepsilon_0 A \delta(\mathbf{r}) dV =
4 \pi \varepsilon_0 A$$",
      "dependency": [
        15
      ]
```

```
    },
    {
      "index": 23,
      "formula": "$$Q = −4\pi \varepsilon_0 A + 4\pi \varepsilon_0 A$$",
      "dependency": [
        21,
        22
      ]
    },
    {
      "index": 24,
      "formula": "$$Q = 0$$",
      "dependency": [
        23
      ],
      "is_final_answer": true
    }
  ]
}
```

# D  DATASET STATISTICS

## D.1  DIFFICULTY ANNOTATION DETAILS

We assign a composite difficulty label by combining LLM-based annotations and structural DAG complexity:

**LLM-Labelled Conceptual and Computational Scores.** Each problem is evaluated along two dimensions using an LLM: $C_1$, which measures *conceptual depth* (the underlying physical principles and modeling complexity), and $C_2$, which captures the *computational burden* (the extent of algebraic or numerical effort required). Both dimensions are rated on a three-level ordinal scale (1–3), with higher values indicating greater complexity.

---

**Prompt for Difficulty Annotation**

You are an experienced Physics Olympiad coach and grader.
Classify Olympiad-level physics problems using TWO dimensions (1–3 each):
C1 Conceptual depth (principles & modeling complexity)
C2 Computation burden (algebra/numeric length, error-prone)

**Rules:**
- Do NOT solve or judge correctness; only estimate difficulty.
- Use the provided SOLUTION only to estimate step depth/concepts.
- Do not use outside tools or knowledge beyond the given text.
- Keep outputs concise.

Output STRICT JSON:

```
{
  "scores": {"C1": 1-3, "C2": 1-3},
  "rationales": {"C1": " <= 20 words", "C2": " <= 20 words"},
  "reasoning": "2-3 concise sentences",
  "confidence": 0.0-1.0
}
```

PROBLEM: {problem}

SOLUTION (only for estimating steps/concepts; do NOT grade correctness): {solution}

---

**Entropy complexity of the solution DAG.** We compute an entropy-based structural complexity by treating the branching at each layer as the search space:

$$e = \sum_{\ell=1}^{\text{Depth}} \log(\text{Width}_\ell).$$

This value is discretized into $C_3 \in \{1, 2, 3\}$ using thresholds $\tau_1, \tau_2$.

**Composite difficulty.** We define a composite score $S = C_1 + C_2 + C_3$, and map it into three difficulty levels: *Easy*, *Medium*, and *Hard*. This composite annotation captures both physics/content difficulty and structural reasoning complexity, providing a stratified view of model performance.

### D.2 DOMAIN CATEGORIZATION DETAILS

The dataset covers a wide range of topics, organized hierarchically into 7 key physics domains and 28 subtopics:

- **Mechanics**: Newtonian Mechanics, Analytical Mechanics, Special Relativity
- **Electromagnetism**: Electrostatics, Magnetostatics and Quasi-Stationary Fields, Electromagnetic Waves
- **Optics**: Geometrical Optics, Wave Optics, Quantum Optics
- **Atomic, Nuclear, and Particle Physics**: Atomic and Molecular Physics, Nuclear Physics, Particle Physics, Other
- **Thermodynamics and Statistical Physics**: Thermodynamics, Statistical Physics
- **Quantum Mechanics**: Basic Principles and One-Dimensional Motions, Central Potentials, Spin and Angular Momentum, Motion in Electromagnetic Fields, Perturbation Theory, Scattering and Transitions, Many-Particle Systems, Other
- **Solid State Physics and Miscellaneous Topics**: Solid State Physics, Relativity, Other

# E    EXPERIMENTAL DETAILS FOR PRISM-PHYSICS

## E.1    EXPERIMENTAL SETUPS

We evaluate a diverse set of 17 leading LLMs, as listed in Table 3. Each model is accessed via its official API using standardized decoding parameters. By default, we set the maximum token output to 8096, `temperature` to 0.0, for all models where these settings are applicable. For reasoning models, the default reasoning effort is chosen as `medium`. Model-specific parameters are specified in the table.

| # | Model | Reasoning | Model Engine Name | Source |
|---|---|---|---|---|
| _Open-source Chat LLMs_ | | | | |
| 1 | Deepseek-V3 DeepSeek-AI (2025) | F | deepseek-v3 | Link |
| 2 | Qwen2.5-72B-Instruct-Turbo Qwen Team (2024) | F | Qwen2.5-72B-Instruct-Turbo | Link |
| 3 | Llama-4-Scout-17B-16E Meta Platforms, Inc. (2025) | F | Llama-4-Scout-17B-16E-Instruct | Link |
| 4 | Llama-3.3-70B-Instruct-Turbo AI (2024) | F | Llama-3.3-70B-Instruct-Turbo | Link |
| _Open-source Reasoning LLMs_ | | | | |
| 5 | Deepseek-R1 DeepSeek-AI (2025) | T | DeepSeek-R1 | Link |
| 6 | GPT-OSS-20B OpenAI (2025d) | T | GPT-OSS-20B | Link |
| 7 | GPT-OSS-120B OpenAI (2025d) | T | GPT-OSS-120B | Link |
| 8 | Qwen3-235B-A22B-Instruct Qwen Team (2025) | T | Qwen3-235B-A22B-Instruct | Link |
| _Proprietary Chat LLMs_ | | | | |
| 9 | Claude-sonnet-4 Anthropic (2025) | T | claude-sonnet-4-20250514 | Link |
| 10 | GPT-4o-mini OpenAI (2024) | F | gpt-4o-mini | Link |
| 11 | GPT-4.1 OpenAI (2025a) | F | gpt-4.1 | Link |
| _Proprietary Reasoning LLMs_ | | | | |
| 12 | Gemini-2.5-Flash Google DeepMind (2025a) | T | gemini-2.5-flash | Link |
| 13 | Gemini-2.5-Pro Google DeepMind (2025b) | T | gemini-2.5-pro | Link |
| 14 | GPT-5 OpenAI (2025b) | Low/Medium/High | gpt-5 | Link |
| 15 | GPT-5-mini OpenAI (2025b) | Low/Medium/High | gpt-5-mini | Link |
| 16 | Grok-4 xAI (2025) | T | grok-4 | Link |
| 17 | o4-mini OpenAI (2025c) | T | o4-mini | Link |
| _Multimodal Large Language Models_ | | | | |
| 18 | Gemini-2.5-Pro Google DeepMind (2025b) | T | gemini-2.5-pro | Link |
| 19 | Gemini-2.5-Flash Google DeepMind (2025a) | T | gemini-2.5-flash | Link |
| 20 | GPT-5 OpenAI (2025b) | Medium | gpt-5 | Link |
| 21 | GPT-5-mini OpenAI (2025b) | Medium | gpt-5-mini | Link |

Table 3: List of LLMs evaluated in our experiments.

**Inference Prompt.**    Below is the prompt we use for inference:

---

**Inference Prompt**

You are a Physics expert. You are going to solve a physics problem and be graded accordingly. Here are some instructions you should follow to make sure your answer is graded correctly: 1. You answer should be written in markdown format. 2. You should provide your key steps and final answer in a clear and concise manner using double-dollar signs for formulas (e.g.,

$$E_0 = mgH + \frac{1}{2}mv_0^2$$

). However, put your definitions or uncrucial steps in single dollar signs (e.g., '$E$ is the energy of the system', or 'according to Newtons second law, $F = ma$'). 3. Use less text and more formulas to explain your reasoning. 4. Your answer should satisfy the format below: **Format Instructions** Here is the problem context: **the problem** Please provide your solution step by step, and then give your final answer. You should try to use the variables given in the problem and avoid using new variables unless necessary. You should strictly follow the formatting requirements introduced in the problem for the final answer.

---

## E.2 ADDITIONAL RESULTS

**Text Models vs. Multimodal Models** We compute the performance gap between multimodal and text-only settings and visualize the differences in Figure 9.

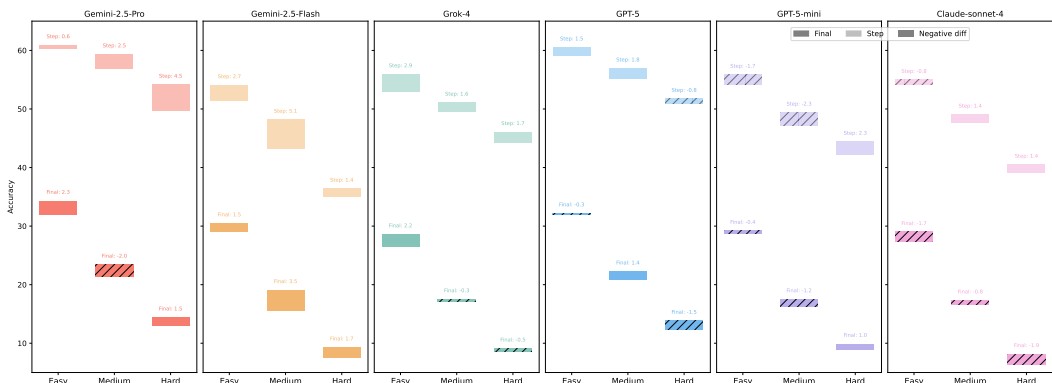

Figure 9: Performance differences between multimodal and text-only settings across models and difficulty levels.

# F EVALUATION FRAMEWORK ANALYSIS DETAILS

## F.1 ANNOTATOR INFORMATION

The majority of the student coauthors of this work have prior experience in Physics Olympiad competitions and are currently pursuing, or about to begin PhD studies in computer science with a focus on AI and LLM research. Almost every member has attained a first prize in regional physics Olympiads, with more than half earning medals at the National Physics Olympiads, including one International Physics Olympiad (IPhO) gold medalist (Rank 10 Globally). This unique combination of competitive physics training and advanced LLM research experience offers a valuable perspective for the development of this work.

## F.2 KENDALL'S TAU-B AND P-TEST

We evaluate the agreement between model-derived scores and human annotations using *Kendall's $\tau_b$ correlation coefficient*, a nonparametric rank-based statistic that extends Kendall's $\tau$ by correcting for ties. Let $\{(x_i, y_i)\}_{i=1}^n$ be a set of paired observations, where $x_i$ represents the score assigned by the model and $y_i$ the corresponding human annotation. Kendall's $\tau_b$ measures the degree to which the rankings induced by $x$ and $y$ agree.

**Definition.** Consider all unordered pairs of distinct indices $(i, j)$ with $1 \leq i < j \leq n$. For each pair, define:

- concordant if $(x_i - x_j)(y_i - y_j) > 0$,
- discordant if $(x_i - x_j)(y_i - y_j) < 0$,
- tied in $x$ if $x_i = x_j$ but $y_i \neq y_j$,
- tied in $y$ if $y_i = y_j$ but $x_i \neq x_j$,
- tied in both if $x_i = x_j$ and $y_i = y_j$.

Let $n_c$ and $n_d$ denote the number of concordant and discordant pairs, respectively. Define

$$n_0 = \tfrac{1}{2}n(n-1), \quad n_1 = \sum_k \tfrac{1}{2}t_k(t_k - 1), \quad n_2 = \sum_l \tfrac{1}{2}u_l(u_l - 1),$$

where $t_k$ is the size of the $k$-th tie group in $x$ and $u_l$ is the size of the $l$-th tie group in $y$. Then Kendall's $\tau_b$ is

$$\tau_b = \frac{n_c - n_d}{\sqrt{(n_0 - n_1)(n_0 - n_2)}}. \tag{2}$$

By construction, $\tau_b \in [-1, 1]$, with $\tau_b = 1$ indicating perfect agreement (all pairs concordant), $\tau_b = -1$ perfect disagreement (all pairs discordant), and $\tau_b = 0$ representing no association beyond what would be expected by chance.

**Statistical significance.** To assess whether the observed correlation is statistically significant, we test the null hypothesis $H_0 : \tau_b = 0$ against the two-sided alternative $H_1 : \tau_b \neq 0$. Two approaches are employed:

1. *Asymptotic test.* Under $H_0$, the sampling distribution of $\tau_b$ is approximately normal for large $n$, with variance given by a closed-form expression that accounts for ties. A standardized statistic $Z = \frac{\tau_b}{\sigma_{\tau_b}}$ is used to compute an asymptotic p-value.

2. *Permutation test.* To avoid reliance on asymptotic approximations, we perform a nonparametric randomization procedure: one ranking (e.g., $y$) is permuted uniformly at random while $x$ remains fixed, and $\tau_b$ is recomputed. Repeating this procedure yields an empirical null distribution for $\tau_b$, from which a p-value is estimated. This approach is robust to small sample sizes and ties.

Together, $\tau_b$ provides a rigorous, tie-adjusted measure of ordinal association, and the combination of asymptotic and permutation-based tests ensures robust inference on the agreement between model predictions and human judgments.

### F.3 FAILURE ANALYSIS FOR EVALUATION FRAMEWORK

We analyzed failure cases from our evaluator and two baselines to understand strengths and limitations. When scoring strictly by formula matching, **causality-aware evaluation is essential**: requiring every reference formula to match is overly rigid and penalizes otherwise correct reasoning. We observe three recurrent failure modes:

- **Contextual vs. literal equivalence.** Two expressions can be equivalent *given the problem context* but not algebraically identical (e.g., re-parameterized integrals or vector identities).

- **Textual answers.** Description- or text-only responses fall outside the scope of strict symbolic matching.

- **Parsing gaps.** Some LaTeX symbols/commands are not reliably recognized as operators, yielding spurious mismatches (e.g., integrals with differentials, vector/tensor notation).

In order to solve these issues, we propose the following roadmap:

1. **Context-aware matching.** We plan to develop a context-sensitive equivalence checker; this is left for future work.

2. **Text evaluation.** We will integrate lightweight LLM "helpers" to assess description-based answers, making the framework more complete while keeping the core scorer deterministic.

3. **Robust parsing.** Despite many fixes, long-tail LaTeX idiosyncrasies remain. We will release the formula matcher first and iteratively expand operator coverage based on community feedback.

# G  ERROR ANALYSIS DETAILS

## G.1  PROMPTS FOR ERROR ANALYSIS

---

**Prompt for Error Analysis**

You are a Physics Olympiad grader. Your task is to analyze a student's solution against a standard solution, using the provided detailed scoring breakdown, and determine the PRIMARY error cause (and optional secondary causes) from the taxonomy below. You do NOT need to align or map steps — the scored expressions already indicate where the solution is correct or incorrect. Focus on WHY the incorrect parts are wrong.

**Error taxonomy (choose labels exactly):**
- DAE: Diagram Analysis Error — incorrect interpretation of diagrams/figures/schematics.
- PTAE: Physics Theorem/Application Error — misuse/misapplication of physical laws/principles.
- MPUE: Modeling & Process Understanding Error — incorrect/incomplete physical model or process understanding.
- CAE: Condition/Assumption Error — invalid/unjustified/misapplied conditions, including boundary/initial conditions.
- VRE: Variable Relationship Error — incorrect relationships between physical quantities (e.g., constraints, kinematic relations).
- DCE: Derivation & Computation Error — algebraic/symbolic manipulation mistakes, arithmetic/sign/substitution errors.
- UDE: Unit/Dimension Error — unit inconsistency or dimensional mismatch.

**General guidance:** 1) Use the scoring breakdown to identify which expressions are incorrect.
2) For the incorrect expressions, determine the earliest fundamental cause from the taxonomy.
3) If multiple causes apply, select ONE primary label and list the rest as secondary.
4) If diagrams are referenced but missing, do not assume their content; judge only from given text/expressions.

Output STRICT JSON (no extra text, no markdown):

```
{
    "primary_error": "DAE|PTAE|MPUE|CAE|VRE|DCE|UDE",
    "secondary_errors": ["DAE|PTAE|MPUE|CAE|VRE|DCE|UDE"],
    "incorrect_expressions":
    [
        "strings of the incorrect student expressions"
    ],
    "related_correct_expressions":
    [
        "strings of correct related student expressions if any"
    ],
    "rationale": "2-5 concise sentences explaining the diagnosis",
    "confidence": 0.0-1.0
}
```

PROBLEM CONTEXT: {`problem`}

STUDENT ANSWER: {`student_answer`}

AUTO-GRADER EQUATION MATCHES: {`matches`}

SCORE: {`score:.3f`} (out of 1.0)

---

## G.2  ERROR TAXONOMY DEFINITIONS

We provide detailed definitions of the seven error categories used in our error analysis:

- **Diagram Analysis Error (DAE)** – incorrect interpretation of diagrams, figures, or schematic representations.

- **Physics Theorem Application Error (PTAE)** – incorrect or inappropriate use of physical laws, theorems, or principles.
- **Modeling and Process Understanding Error (MPUE)** – incorrect or incomplete construction of the physical model, including misunderstanding of the physical process being analyzed.
- **Condition or Assumption Error (CAE)** – invalid, unjustified, or misapplied physical conditions, including boundary or initial conditions.
- **Variable Relationship Error (VRE)** – incorrect establishment or use of relationships between physical quantities.
- **Derivation and Computation Error (DCE)** – incorrect algebraic manipulation, symbolic transformation, arithmetic mistakes, sign errors, or incorrect numerical substitutions.
- **Unit Dimension Error (UDE)** – inconsistency in physical units or failure to maintain dimensional correctness.

## G.3 MODEL FAILURE SOLUTION EXAMPLES

---

**PRISM-PHYSICS Failure Example: Condition or Assumption Error**

**Problem:** A particle of mass $m$ moves under the influence of an attractive central force $f(r)$. The potential energy associated with this force is $V(r)$, defined so that $f(r) = -\frac{dV}{dr}$. The angular momentum of the particle is $J$. The effective potential is denoted by $V^*(r)$. The total mechanical energy is $E$. The instantaneous radial coordinate of the particle is $r$, and the radial velocity is $\dot{r} = \frac{dr}{dt}$. The radius of the circular orbit is $r_0$. For the stability analysis, the student should write their answer for the maximum $n$ as $ans = ...$, where $n$ is the exponent in the force law $f(r) = -\frac{K}{r^n}$, with $K$ a positive constant.
(c) Determine the maximum value of $n$ for which the circular orbit is stable under the force law $f(r) = -\frac{K}{r^n}$, where $K$ is a positive constant.

**GroundTruth Solution:**

For $f(r) = -\frac{K}{r^n}$, we have

$$\frac{\partial f}{\partial r} = n\frac{K}{r^{n+1}}$$

From the equilibrium condition,

$$\frac{\partial V}{\partial r} = \frac{J^2}{mr^3}$$

Also, $\frac{\partial V}{\partial r} = -f(r)$, so

$$-f(r) = \frac{J^2}{mr^3}$$

which gives

$$\frac{K}{r^n} = \frac{J^2}{mr^3}$$

Thus,

$$J^2 = mKr^{3-n}$$

Insert this value for $J^2$ into the stability condition:

$$\frac{3J^2}{mr^4} - \frac{\partial f}{\partial r} > 0$$

Substitute $J^2 = mKr^{3-n}$ and $\frac{\partial f}{\partial r} = n\frac{K}{r^{n+1}}$

$$\frac{3K}{r^{n+1}} - \frac{nK}{r^{n+1}} > 0$$

So

$$3 - n > 0$$

Therefore,
$$\text{ans} = n < 3$$

**Student Solution Snippet:**

(c): Maximum Value of n For the force law $f(r) = -\frac{K}{r^n}$:

$$\frac{\partial f}{\partial r} = \frac{nK}{r^{n+1}}$$

Substituting into the stability condition:

$$\frac{nK}{r^{n+1}} + \frac{3(-K/r^n)}{r} < 0$$

$$\frac{nK}{r^{n+1}} - \frac{3K}{r^{n+1}} < 0$$

$$\frac{K(n-3)}{r^{n+1}} < 0$$

Since $K > 0$ and $r^{n+1} > 0$, we need:

$$n - 3 < 0$$

$$n < 3$$

The maximum value of $n$ for stability is therefore $n = 3$.

$$\text{ans} = 3$$

**Error Analysis Output:**

```
{
 "primary_error": "CAE",
 "incorrect_expressions":  [

   "The maximum value of n for stability is therefore n = 3.",
   "ans= 3"

 ],

 "related_correct_expressions":  [
```

$$n < 3,$$

$$\frac{K(n-3)}{r^{n+1}} < 0$$

```
 ],

 "rationale":  "All algebra and the stability condition
were derived correctly, yielding n < 3.  The student
then incorrectly interpreted this strict inequality as
allowing n = 3 as the maximum stable value.  This is an
assumption/boundary-condition mistake:  n = 3 is the marginal
(neutral) case, not a stable one, so the conclusion ans = 3 is
unjustified."

}
```

---

**PRISM-PHYSICS Failure Example: Diagram Analysis Error**

**Problem:** An aquarium is being filled with water of refractive index $n > 1$. A stationary fish inside the aquarium looks vertically upward through the rising surface of the water at a stationary monochromatic plane wave light source outside the aquarium. The water surface rises upward at constant speed $V$ relative to both the fish and the source, while the rest of the water and the fish are stationary with respect to the aquarium. The fish observes the incoming light through the moving water interface as the aquarium fills. Use the following variable definitions:- $n$ is the refractive index of water- $V$ is the upward speed of the rising water surface with respect to both fish and light source- $c$ is the speed of light in vacuum- $u$ is the frequency of light as emitted by the source in vacuum- $u'$ is the frequency of light as observed by the fish inside the water- $u$ is the speed of light in water as measured by the fish- $\lambda_w$ is the wavelength of light in waterExpress your answer for $u'$ in terms of $u$, $V$, $n$, and $c$. Also state, using an equation, whether $u'$ is less than, greater than, or equal to $u$.

**GroundTruth Solution:**

The speed of light in water as measured by the fish is

$$u = \frac{c}{n} - V\left(1 - \frac{1}{n^2}\right)$$

The wavelength of the light in water is

$$\lambda_w = \frac{c}{n\nu}$$

The observed frequency by the fish is

$$\nu' = \frac{u}{\lambda_w}$$

Substituting for $u$ yields

$$\nu' = \frac{\frac{c}{n} - V\left(1 - \frac{1}{n^2}\right)}{\lambda_w}$$

Substituting for $\lambda_w$ yields

$$\nu' = \frac{\frac{c}{n} - V\left(1 - \frac{1}{n^2}\right)}{\frac{c}{n\nu}}$$

This simplifies to

$$\nu' = \left(\frac{c}{n} - V\left(1 - \frac{1}{n^2}\right)\right)\frac{n\nu}{c}$$

Expanding the terms gives

$$\nu' = \nu - V\left(1 - \frac{1}{n^2}\right)\frac{n\nu}{c}$$

For $n > 1$ and $V > 0$,

$$\nu' < \nu$$

Therefore, the fish observes the light source to be red-shifted.

**Student Solution Snippet:**

We analyze the scenario using the Doppler effect for light at a moving interface. The light source is stationary in air (refractive index $\approx 1$), emitting light of frequency $u$ and speed $c$. The water surface rises at speed $V$ upward, and the fish is stationary in water of refractive index $n$. The fish observes the frequency $u'$ of light that has entered the water.

The speed of light in water as measured by the fish is

$$u = \frac{c}{n}.$$

The incident light in air approaches the moving water surface. Since the interface is moving *toward* the source (upward, while light comes from above), the frequency perceived just after crossing into water is Doppler shifted.

Let the incident plane wave in air be

$$E_i = E_0 \cos(\omega t - kz), \quad \omega = 2\pi u, \quad k = \frac{\omega}{c}.$$

At the interface $z = Vt$, the phase is

$$\phi_i(t) = \omega t - k(Vt) = \omega t \left(1 - \frac{V}{c}\right).$$

The transmitted wave in water is

$$E_t = E_1 \cos(\omega' t - k'z), \quad k' = \frac{n\omega'}{c}.$$

At $z = Vt$, its phase is

$$\phi_t(t) = \omega' t - k'Vt.$$

Continuity of phase requires $\phi_i(t) = \phi_t(t)$, hence

$$\omega \left(1 - \frac{V}{c}\right) = \omega' - k'V.$$

Substituting $k' = \frac{n\omega'}{c}$ gives

$$\omega' \left(1 - \frac{nV}{c}\right) = \omega \left(1 - \frac{V}{c}\right).$$

Thus,

$$\omega' = \omega \frac{1 - \frac{V}{c}}{1 - \frac{nV}{c}}.$$

Equivalently, in terms of frequency:

$$u' = u \frac{1 - \frac{V}{c}}{1 - \frac{nV}{c}} = u \frac{c - V}{c - nV}.$$

Now compare $u'$ and $u$:

$$u' - u = u \left(\frac{c - V}{c - nV} - 1\right) = u \frac{(n-1)V}{c - nV}.$$

Since $n > 1$, $V > 0$, and $c - nV > 0$ (non-relativistic case), we have $u' - u > 0$. Therefore,

$$u' > u.$$

*Final result:*

$$u' = u \frac{1 - \frac{V}{c}}{1 - \frac{nV}{c}}, \qquad u' > u.$$

**Error Analysis Output:**

```
{
  "primary_error": "DAE",
```

```
"incorrect_expressions": [
```

$$E_i = E_0 \cos(\omega t - kz), \quad \omega = 2\pi\nu, \quad k = \frac{\omega}{c},$$

$$\phi_i(t) = \omega t - k(Vt) = \omega t - \frac{\omega}{c} V t = \omega t \left(1 - \frac{V}{c}\right),$$

$$\omega' - k'V = \omega \left(1 - \frac{V}{c}\right),$$

$$\omega' = \omega \frac{1 - \frac{V}{c}}{1 - \frac{nV}{c}},$$

$$\nu' = \nu \frac{1 - \frac{V}{c}}{1 - \frac{nV}{c}}$$

```
],
"related_correct_expressions": [
```

$$u = \frac{c}{n},$$

$$k' = \frac{\omega'}{u} = \frac{n\omega'}{c},$$

Continuity of phase at the moving interface: $\phi_i(z = Vt) = \phi_t(z = Vt)$

```
],
"rationale": "The student wrote the incident wave as
```
$\omega t - kz$, which corresponds to a wave propagating upward; for light coming from above toward the interface (downward propagation) the sign of the spatial term is wrong. This incorrect sign gives the wrong expression for the incident phase at $z = Vt$ and therefore leads to the incorrect algebraic relation and final formula for $\nu'$. The use of phase continuity and $k' = n\omega'/c$ and $u = c/n$ are otherwise appropriate, so the error is an incorrect interpretation of the wave propagation direction (diagram/sign error)."
```
}
```

---

**PRISM-PHYSICS Failure Example: Derivation and Computation Error**

**Error Analysis Output:**

```
{
  "primary_error": "DCE",
  "incorrect_expressions": [
```

$$\frac{\partial L}{\partial r} = mr\dot{\theta}^2 + \frac{k}{r^2} - \frac{k'}{r^3},$$

$$m\ddot{r} = mr\dot{\theta}^2 + \frac{k}{r^2} - \frac{k'}{r^3}$$

```
],

  "related_correct_expressions": [
```

$$\frac{\partial L}{\partial r} = mr\dot{\theta}^2 - \frac{k}{r^2} + \frac{k'}{r^3},$$

$$m\ddot{r} = mr\dot{\theta}^2 - \frac{k}{r^2} + \frac{k'}{r^3}$$

```
],

  "rationale": "The student made a sign/algebra error when
```
differentiating the potential terms in L with respect to r, flipping the signs of the k and k' contributions. This is a

```
pure derivation/computation mistake (not a misapplication of
physics), since the correct radial equation has the opposite
signs and follows from the correctly differentiated Lagrangian
or directly from Newton's form.  The later orbit derivation
uses the correct form of F(r), so the error is localized to the
symbolic differentiation step."

}
```

**PRISM-PHYSICS Failure Example: Modeling and Process Understanding Error**

**Problem:** A particle of mass $m$ moves under a restoring force $-Kx$ and a resistive force $-Rv$, where $x$ is the displacement from equilibrium and $v$ is the particle's velocity. Here, $m$ is the mass of the particle, $K$ is the spring constant, $R$ is the damping coefficient, $x$ is the displacement from equilibrium, and $v$ is the velocity. For fixed $K$ and arbitrary initial conditions, find the value $R = R_c$ that gives the most rapid approach to equilibrium. Also, state whether it is possible to choose initial conditions (other than $x = 0$ and $v = 0$) such that the approach to equilibrium is more rapid for $R > R_c$ or $R < R_c$. Specifically, $R_c$ should be defined in terms of $K$ and $m$. Express the rate of approach and clearly specify all variables throughout. The final answer for the most rapid approach value should be given as $R_c =$ in terms of $K$ and $m$.

**GoundTruth Solution:**

The equation of motion for the particle can be written as

$$m\frac{d^2x}{dt^2} + R\frac{dx}{dt} + Kx = 0$$

......

Thus, for certain special (non-generic) initial conditions, the approach to equilibrium may be even faster than in the critically damped case.If $R < R_c$, the roots are complex:

$$\alpha = \frac{-R \pm i\sqrt{R_c^2 - R^2}}{2m}$$

The general solution is

$$x = A\exp\left(-\frac{Rt}{2m}\right)\cos\left(\frac{\sqrt{R_c^2 - R^2}}{2m}t\right) + B\exp\left(-\frac{Rt}{2m}\right)\sin\left(\frac{\sqrt{R_c^2 - R^2}}{2m}t\right)$$

The rate of decay towards equilibrium is determined by the exponential factor

$$\frac{R}{2m} < \frac{R_c}{2m}$$

Thus, the approach to equilibrium is always slower than for critical damping when $R < R_c$. In summary, the value of $R$ that gives the most rapid approach to equilibrium for arbitrary initial conditions is

$$R_c = 2\sqrt{Km}$$

For $R > R_c$, in principle, one can choose initial conditions such that the approach is even more rapid, but this occurs only for special initial conditions (such as setting $A = 0$ in the overdamped general solution). In general, for arbitrary initial conditions, critical damping ($R = R_c$) gives the most rapid approach. For $R < R_c$, the approach to equilibrium is always slower than for critical damping.

**Student Solution Snippet:**

The equation of motion for a damped harmonic oscillator is:

$$m\frac{d^2x}{dt^2} = -Kx - R\frac{dx}{dt}$$

Rearranging to standard form:

$$\frac{d^2x}{dt^2} + \frac{R}{m}\frac{dx}{dt} + \frac{K}{m}x = 0$$

Let $\omega_0^2 = \frac{K}{m}$ and $2\gamma = \frac{R}{m}$, so the equation becomes:

$$\frac{d^2x}{dt^2} + 2\gamma\frac{dx}{dt} + \omega_0^2 x = 0$$

The characteristic equation is:

$$r^2 + 2\gamma r + \omega_0^2 = 0$$

The roots are:

$$r = -\gamma \pm \sqrt{\gamma^2 - \omega_0^2}$$

The behavior depends on the discriminant $\gamma^2 - \omega_0^2$:

*Case 1: Underdamped ($\gamma < \omega_0$ or $R < 2\sqrt{Km}$)*

$$x(t) = e^{-\gamma t}(A\cos(\omega_d t) + B\sin(\omega_d t))$$

where $\omega_d = \sqrt{\omega_0^2 - \gamma^2}$.

*Case 2: Critically damped ($\gamma = \omega_0$ or $R = 2\sqrt{Km}$)*

$$x(t) = e^{-\gamma t}(A + Bt)$$

*Case 3: Overdamped ($\gamma > \omega_0$ or $R > 2\sqrt{Km}$)*

$$x(t) = Ae^{r_1 t} + Be^{r_2 t}$$

where $r_1 = -\gamma + \sqrt{\gamma^2 - \omega_0^2}$ and $r_2 = -\gamma - \sqrt{\gamma^2 - \omega_0^2}$. For the most rapid approach to equilibrium, we need to minimize the time for $x(t) \to 0$.

In the underdamped case, the envelope decays as $e^{-\gamma t}$. In the critically damped case, the decay is $e^{-\gamma t}$. In the overdamped case, the slowest decay rate is determined by $r_1 = -\gamma + \sqrt{\gamma^2 - \omega_0^2}$. Since $\sqrt{\gamma^2 - \omega_0^2} > 0$ when $\gamma > \omega_0$, we have $|r_1| < \gamma$, making overdamped systems slower than critically damped.

For underdamped systems, the envelope decay rate is $\gamma = \frac{R}{2m}$. For critically damped systems, the decay rate is also $\gamma = \frac{R}{2m}$. However, the critically damped case has no oscillations and provides the fastest approach without overshoot. Therefore, the critical damping condition gives the most rapid approach:

$$R_c = 2\sqrt{Km}$$

Regarding initial conditions: For any given $R$, the approach rate is determined by the system parameters, not the initial conditions. The initial conditions only affect the amplitude coefficients $A$ and $B$, but not the fundamental decay rates. Therefore, it is **not possible** to choose initial conditions to make $R > R_c$ or $R < R_c$ approach equilibrium more rapidly than $R = R_c$.

$$R_c = 2\sqrt{Km}$$

**Error Analysis Output:**

```
{
  "primary_error": "MPUE",
 "incorrect_expressions":  [ "Therefore, it is not possible
to choose initial conditions to make R  >  R_c or R  <  R_c
approach equilibrium more rapidly than R  =  R_c.", "However,
the critically damped case has no oscillations and provides the
fastest approach without overshoot." ],
```

```
  "related_correct_expressions":  [
```

$$m\frac{d^2x}{dt^2} = -Kx - R\frac{dx}{dt},$$

$$R_c = 2\sqrt{Km},$$

$$r = -\gamma \pm \sqrt{\gamma^2 - \omega_0^2},$$

Underdamped:   $x(t) = e^{-\gamma t}(A\cos(\omega_d t) + B\sin(\omega_d t))$,

Critically damped:   $x(t) = e^{-\gamma t}(A + Bt)$,

Overdamped:   $x(t) = Ae^{r_1 t} + Be^{r_2 t}$,   $r_1 = -\gamma + \sqrt{\gamma^2 - \omega_0^2}$

```
  ],
  "rationale":  "The student correctly derived the equation
of motion , root structure, and critical damping value R_c,
but then asserted that initial conditions cannot alter which
R gives the fastest approach.  That is a modelling process
misunderstanding:  transient approach time can depend on
initial conditions (e.g.  certain initial displacements or
velocities can make overdamped underdamped responses reach
zero sooner for some measures).  The blanket statement that
critical damping is always fastest for every initial condition
is therefore incorrect."
}
```

---

### PRISM-PHYSICS Failure Example: Physics Theorem Application Error

**Problem**: A bucket of mass $M$ (when empty) initially at rest and containing a mass of water is being pulled up a well by a rope exerting a steady force $P$ The water is leaking out of the bucket at a steady rate such that the bucket is empty after a time $\tau$ .Find the velocity of the bucket at the instant it becomes empty.

**GroundTruth Solution** :

Let the total mass of the bucket and water at time $t$ be $M'$.

$$M' = M + m - \frac{mt}{\tau}$$

The equation of motion for the bucket (plus remaining water) is

$$M'\frac{dv}{dt} = P - M'g$$

Rearranging gives

$$dv = \left(\frac{P}{M + m - \frac{m}{\tau}t} - g\right) dt$$

The velocity $v$ of the bucket at time $t = \tau$ (when empty) starting from rest is

$$v_{empty} = \int_0^\tau \frac{P dt}{M + m - \frac{m}{\tau}t} - g\tau$$

Evaluating the integral yields

$$v_{empty} = \frac{P\tau}{m} \ln\left(\frac{M + m}{M}\right) - g\tau$$

where $v_{empty}$ is the velocity of the bucket at the instant it becomes empty.

**Student Solution Snippet:**

I need to find the velocity of the bucket when it becomes empty at time $t = \tau$. Setting up the problem The mass of water decreases linearly from $m$ to $0$ over time $\tau$:

$$m_{water}(t) = m(1 - \frac{t}{\tau})$$

The total mass of the system at time $t$ is:

$$M_{total}(t) = M + m(1 - \frac{t}{\tau})$$

The rate of mass loss is:

$$\frac{dm_{water}}{dt} = -\frac{m}{\tau}$$

*Applying Newton's Second Law*
Since the leaked water has zero velocity relative to the bucket, when it leaves the bucket, it has the same velocity $v$ as the bucket. The forces acting on the system are:
- Upward force: $P$ - Weight: $(M + m_{water}(t))g$
Using Newton's second law for variable mass systems:

$$P - (M + m(1 - \frac{t}{\tau}))g = (M + m(1 - \frac{t}{\tau}))\frac{dv}{dt} + v\frac{dm_{water}}{dt}$$

......

**Error Analysis Output:**

```
\begin{verbatim}
{
  "primary_error": "PTAE",
  "incorrect_expressions":  [
```

$$P - (M + m(1 - \tfrac{t}{\tau}))g = (M + m(1 - \tfrac{t}{\tau}))\tfrac{dv}{dt} + v\tfrac{dm_{water}}{dt}$$

```
],

  "related_correct_expressions":  [
```

$dm_{water}/dt = -m/\tau$   (student's expression, correct),
$(M + m(1 - t/\tau))\frac{dv}{dt} = P - (M + m(1 - t/\tau))g,$
(since escaping water has zero velocity relative to the bucket, the thrust term vanishes)
$v_{empty} = \frac{P\tau}{m}\ln\left(\frac{M+m}{M}\right) - g\tau$

```
 ],
"rationale":  "The student misapplied the variable-mass form of
Newton's second law:  the net external force equal to d/dt(Mv)
and thus kept a v dM/dt term on the RHS (or equivalently
omitted the momentum-flux term on the RHS), which is incorrect
bookkeeping for mass leaving with zero velocity relative to
the bucket.  For escaping water with velocity equal to the
bucket, the relative-velocity term vanishes and the correct
ODE is M(t) dv/dt = P - M(t) g, leading to a different integral
(logarithmic) result.  All subsequent algebra and the final
numeric expression therefore follow from this incorrect
application."
}
```

---

**PRISM-PHYSICS Failure Example: Unit Dimension Error**

**Error Analysis Output:**

```
{
  "primary_error": "UDE",
 "incorrect_expressions":  [
```

$A = \frac{1}{\lambda^2 \sigma_t^2 T}$,

$N = \frac{1}{\lambda^3 \sigma_t^2 T}$,

$x = \frac{12}{10^{-12} N_A} \cdot \frac{T_{1/2}^3}{(\ln 2)^3 \sigma_t^2 T} \cdot e^{\lambda t}$,

$x = \frac{12 \cdot (5730)^3}{10^{-12} \cdot 6.023 \times 10^{23} \cdot (\ln 2)^3 \cdot 2500} \cdot e^{(\ln 2) \cdot 5000/5730}$,

$\lambda^2 \sigma_t^2 T A^2 - A - A_B = 0$,

$A = \frac{1 + \sqrt{1 + 4\lambda^2 \sigma_t^2 T A_B}}{2\lambda^2 \sigma_t^2 T}$,

$N = \frac{1 + \sqrt{1 + 4\lambda^2 \sigma_t^2 T A_B}}{2\lambda^3 \sigma_t^2 T}$,

$x = \frac{12}{10^{-12} N_A} \cdot \frac{1 + \sqrt{1 + 4\lambda^2 \sigma_t^2 T A_B}}{2\lambda^3 \sigma_t^2 T} \cdot e^{\lambda t}$

```
 ],

 "related_correct_expressions":  [
```

$A = \lambda N$,

$\frac{dA}{dt} = -\lambda A$,

$\lambda = \frac{\ln 2}{T_{1/2}}$,

$N = N_0 e^{-\lambda t}$,

$N_0 = 10^{-12} \frac{x N_A}{12}$

```
 ],

 "rationale":  "The student mixed time units:  λ and σ_t are in
years while the counting time T was used in hours, so formulas
combining λ and T (e.g.  A = 1/(λ²σ_t²T) and subsequent N and x
expressions) are dimensionally inconsistent.  The algebraic
manipulations themselves are otherwise coherent, but the unit
mismatch renders the numerical/physical results incorrect.  The
same unit inconsistency propagates into the background-case
quadratic and its solutions.",
}
```

---

**PRISM-PHYSICS Failure Example: Variable Relationship Error**

**Error Analysis Output:**

```
{
  "primary_error": "VRE",
 "incorrect_expressions":  [
```

$d = \theta(R + h)$,

$d = 1.22 \frac{\lambda}{D}(R + h)$,

$d = 1.22 \times \frac{1}{1000} \times 4.217 \times 10^7$,

$d \approx 5.145 \times 10^4 \, \text{m}$,

$d = 51450 \, \text{m}$

```
  ],

 "related_correct_expressions":  [
```
$$\theta \approx 1.22 \tfrac{\lambda}{D},$$
$$R + h = \left(\tfrac{GM}{\omega^2}\right)^{1/3},$$
$$h = 3.580 \times 10^7 \,\mathrm{m}$$
```
  ],

 "rationale":  "The student used R + h (distance from Earth's
center to the satellite) as the propagation distance for the
beam instead of the correct path length from satellite to
ground (the height h).  That is an incorrect relationship
between physical quantities (distance to apply diffraction).
The diffraction formula and numerical algebra are otherwise
applied correctly, so the error is conceptual about which
length variable to use.",
}
```

