# OpenReview forum: "PRISM-Physics: Causal DAG-Based Process Evaluation for Physics Reasoning"
_ICLR.cc/2026/Conference — ICLR 2026 Poster_

### Official Review · Reviewer_4yRy · 2025-10-25

**Soundness:** 3
**Presentation:** 3
**Contribution:** 3
**Rating:** 6
**Confidence:** 3

**Summary:**

This paper introduces PRISM-PHYSICS, a benchmark and a process-level evaluation framework that encodes physics solutions as DAGs and employs rule-based symbolic equivalence checking for reliable, fine-grained scoring.

**Strengths:**

1. A large-scale benchmark of competition-level physics problems with carefully curated, DAG-structured solutions.
2. A DAG-based scoring policy that explicitly models causal dependencies among formulas, enabling fine-grained and interpretable process-level evaluation.
3. A fully rule-based symbolic formula equivalence checker to reliably validate diverse mathematical expressions, ensuring consistent comparison across alternative formulations and eliminating reliance on heuristic LLM-as-judge scoring.

**Weaknesses:**

1. My main concern is that Figures 1 through 5 are very unclear, and even when enlarged twice, they are still hard to read. These figures should ideally be the most direct representation of the data analysis in this study, PRISM-PHYSICS. I hope the authors can improve the clarity of these figures.
2. Figure 1 appears on page 2, but there is no corresponding content on the first two pages. Is its placement here inappropriate? Additionally, Figure 1 is not referenced anywhere in the text. The same issue applies to Figure 2.
3. I feel that the originality of the article is somewhat limited. Could the authors provide an explanation of the connection between the challenges presented in this study and the research methods? At the moment, the challenges and methods do not seem to align well.
4. Regarding the analysis of the experimental section, I hope the authors can summarize it more clearly. The current summary of the experiments is not very clear.
5. How was the step level notation done? I would appreciate clarification on this point.

**Questions:**

Refer to the weaknesses.

---

> ### Author Response · Authors · 2025-11-20
>
> We sincerely thank the reviewer for recognizing the value of our DAG-structured benchmark, the interpretability of our DAG-based scoring policy, and the reliability of our rule-based symbolic equivalence checker. We appreciate the positive feedback and address the remaining concerns below.
>
> ### `W1 Regarding Figures 1–5 clarity`
>
> Thank you for pointing this out. We apologize for the earlier oversight. In the revised version, we have improved the resolution, enlarged the labels, and simplified the layout to ensure that all figures are clearly readable.
>
> ### `W2 Regarding placement and referencing of Figures 1 and 2`
>
> We apologize for the confusion. We agree that the original placement may have been unclear. In the revised version, we have added explicit references to Figures 1 and 2 in the main text.
>
> We keep Figure 1 at the beginning because it provides a high-level overview of overall performance, helping readers quickly grasp the dataset and general model behavior before diving into detailed analysis.
>
> Figure 2 summarizes the dataset’s difficulty and category distribution and illustrates a representative DAG example. Presenting them early helps readers establish a clear global understanding before engaging with the detailed formulation and experiments.
>
> ### `W3 Alignment Between Challenges and Methods`
> We appreciate the reviewer’s comment and would like to clarify the connection between the challenges we identify and the methods we propose.
>
> The core challenges in evaluating physics reasoning are:
>
> 1. the lack of fine-grained supervision or principled partial credit in existing benchmarks,
>
> 2. the inherently non-linear, multi-branch structure of physics derivations, and
>
> 3.  the unreliability of LLM-as-judge approaches for symbolic and physical correctness.
>
> Our methods are designed specifically to address these issues:
>
> **(1) DAG-structured solutions directly address the structural challenge (Challenge 2) and support principled partial credit (Challenge 1).**
>
> Physics reasoning is rarely a single sequential chain; derivations involve branching dependencies (e.g., simultaneously using energy conservation and a geometric relation).
> A DAG explicitly represents the branching dependencies of a physics derivation and provides the structural foundation on which meaningful partial credit can be assigned, something linear CoT-style or index-based scoring cannot capture.
>
>
> **(2) Ancestor-Closure Scoring specifically resolves the partial-credit challenge (Challenge 1).**
>
> A long-standing issue in physics evaluation is the inability to assign principled partial credit across diverse solution pathways. Ancestor-Closure Scoring provides a mathematically grounded way to score partially correct reasoning, aligned with how human graders evaluate physics solutions.
>
> **(3) The rule-based symbolic equivalence checker addresses (Challenge 3).**
>
> LLM-as-judge struggles with symbolic manipulation, dimensional consistency, and algebraic equivalence. Instead of relying on LLM-as-judge to decide whether a formula is “close enough,” we use a deterministic rule-based checker to normalize and compare expressions, ensuring consistent treatment of algebraic variants and eliminating stochastic judgment at evaluation time.
>
>
> Together, these components form a cohesive solution: the DAG captures the non-linear structure of physics reasoning, Ancestor-Closure Scoring uses this structure to define principled partial credit, and the symbolic checker ensures that correctness decisions are reliable. The methods are therefore directly motivated by and tightly aligned with the challenges we aim to resolve.

---

> > ### Comment · Reviewer_4yRy · 2025-11-24
> > **Supplement of Comments**
> >
> > Thank you for the detailed response. I appreciate the effort the authors have made in addressing the previous comments; however, my main concerns remain unresolved.
> >
> > First, regarding the motivation and challenges of the study, the current explanation still lacks originality. The motivation should clearly articulate the specific limitations of existing datasets in evaluating LLMs and how the proposed benchmark effectively addresses these gaps. Similarly, in the Introduction, the discussion of limitations does not highlight any concrete technical challenges; it remains at a high-level and does not convincingly justify the need for this study.
> >
> > Second, for Figures 1–5, although the authors mention in the rebuttal that larger and clearer versions have been added to the appendix, the figures in the main text remain unclear. I do not agree that placing clear figures only in the appendix is sufficient. The main text should maintain clarity and completeness. For example, Figures 1 and 2 could be simplified to convey their essential meaning more clearly, rather than prioritizing comprehensiveness or volume of information at the expense of readability. I encourage the authors to reconsider the presentation of these figures.
> >
> > I hope the authors can further revise the manuscript to adequately address the above issues.

---

> > > ### Author Response · Authors · 2025-11-25
> > >
> > > Dear reviewer,
> > >
> > > We were surprised and sorry to see the score lowered from 6 to 4. We are grateful for the additional comments, but it would be extremely helpful if you could clarify which specific points remain unresolved after our detailed rebuttal. Without concrete concerns, it is difficult for us to meaningfully revise the manuscript. Below, we would like to further clarify and address the stated issues:
> > > ### `On the motivation and the alignment between challenges and methods`
> > > We believe there may be some misunderstanding here. Both in the original introduction and in the rebuttal, we explicitly articulated the limitations of existing benchmarks and the technical challenges our framework is designed to address.
> > > **For instance, the introduction clearly states:**
> > >
> > > 1. ***Existing physics benchmarks** rely primarily on multiple-choice or short-answer formats* **(around line 44).**
> > >
> > > 2. ***Most evaluate** only the final answer and rely on LLM-as-judge scoring which is prone to hallucinations, prompt sensitivity, and inconsistent grading.* **(line 45- 46).**
> > >
> > > 3. ***While some recent work** has made initial attempts at step-by-step scoring, these approaches often rely on strong assumptions such as strictly linear step ordering or shallow expression matching that limit the validity and generalizability of the framework.*   **(line 48).**
> > >
> > > These also map directly to **the three concrete challenges we summarized in the previous rebuttal**:
> > >
> > > 1. lack of principled partial-credit evaluation
> > >
> > > 2. the inherently non-linear, multi-branch structure of physics derivations, and
> > >
> > > 3. the unreliability of LLM-as-judge approaches for symbolic or physical correctness.
> > >
> > > And following that, the introduction explains **how our methods effectively address these challenges and gaps**:
> > >
> > > 1. ***process-level evaluation framework** using DAG-structured reasoning* **(around line 53)**
> > >
> > > 2. ***the DAG explicitly encodes causal dependencies*** **(around line 69)**, … *enabling fine-grained and interpretable process-level evaluation*  **(around line 76)**
> > >
> > > 3. *a fully rule-based symbolic equivalence checker **ensures deterministic correctness*** **(around line 71, around line 77-78).**
> > >
> > > In the rebuttal, we further clarified:
> > > 1. how DAG structure **deals with branching dependencies** (linear CoT-style scoring cannot represent),
> > >
> > > 2. **Ancestor Scoring policy solves the partial-credit issue**, and
> > >
> > > 3. **a rule-based equivalence checker eliminates LLM judgment noise**.
> > >
> > > Also, we have provided a more detailed discussion of the limitations of previous work in Section 2 (Related Work). We appreciate your emphasis on the Introduction section, but we believe that the high-level description in the Introduction is already intuitive and clear, while Section 2 offers the appropriate place for a more detailed, fine-grained comparison. **We have also updated the Introduction** to further clarify the motivation and strengthen the alignment between the identified challenges and our methodological contributions.
> > >
> > > If any part of this explanation is still unclear, we would be grateful for specific guidance(e.g., which point felt unconvincing, or what concrete additions would better address your concerns). We are more than willing to expand or restructure the introduction, but we would appreciate more actionable feedback to ensure the revision aligns with your expectations.

---

> > > > ### Author Response · Authors · 2025-11-25
> > > >
> > > > ### `Figure Clarity.`
> > > >
> > > > Thank you for the follow-up comment. We take figure clarity very seriously, and in the revised version we have made substantial changes that directly address your concerns.
> > > >
> > > > Regarding your current comment about ”convey essential meaning more clearly rather than prioritizing comprehensiveness.” **We restructured and simplified** Figure 1 and 2:
> > > >
> > > > Figure 1 has been simplified to **show only selected representative models**, making the main-text visualization much clearer. The full set of models is now included in the appendix, while the main figure provides a concise high-level overview.
> > > >
> > > > Figure 2 has been **split into two focused components**:
> > > >
> > > > (1) A data example with the proposed DAG structure, **placed early to give readers an immediate high-level intuition**;
> > > >
> > > > (2) the dataset distribution, **moved to the data annotation section where it fits naturally and can be shown at larger scale**.
> > > >
> > > > Regarding your earlier comment on enlarging Figures 1–5 twice was still unclear, we have added high-resolution, enlarged versions of all figures in the appendix so that readers who prefer printing or paper viewing can examine them clearly. For readers viewing the PDF digitally,**the figures in the main text can be zoomed losslessly (up to 500%) and remain fully clear and readable.**
> > > >
> > > > Regarding your earlier comment about the placement and missing references for Figures 1 and 2, we have added explicit references in the main text and clarified their intended role. As a benchmark paper, we intentionally place these two figures early to **provide a high-level overview of model performance** (Figure 1) and **a representative DAG example** (Figure 2), helping readers **form a global understanding** before engaging with the detailed formulations and experiments.
> > > >
> > > > These restructuring makes both parts more readable and visually clear. All references in the main text have also been updated accordingly. We hope these revisions effectively address your suggestions and concerns regarding figure clarity, and we welcome any further feedback or constructive suggestions.
> > > >
> > > > ## **Summary**
> > > > Overall, we are grateful for your further comments, and we hope our responses sufficiently address your concerns regarding: **(1)** *The motivation clearly identifies gaps in existing physics datasets and explains how our benchmark addresses them*, **(2)** *Figures convey essential meaning more clearly rather than prioritizing comprehensiveness* and also the previous concern on **(3)** *enlarging Fig1-5 twice is still unclear* **(4)** *Placement and missing reference issue for Fig1-2*.
> > > >
> > > > We would greatly appreciate more specific guidance on where further improvements are needed, and which aspects of our revisions motivated the lowered score. If our clarifications resolve the issues you identified, we would sincerely appreciate your reconsideration of the evaluation. Thank you!

---

> > > > > ### Comment · Reviewer_4yRy · 2025-11-26
> > > > > **Thanks for your explanations.**
> > > > >
> > > > > The author has addressed most of my concerns. Accordingly, I will raise my score.

---

> ### Author Response · Authors · 2025-11-20
>
> ### `W4 Clarity and Organization of Experimental Results`
> Thank you for this valuable feedback. We have added a new subsection (6.2) that provides a concise one-sentence summary for each experiment to make the overall results clearer and more organized. We also put the content here for your reference.
>
> ```
> We conduct a comprehensive set of experiments, including overall performance evaluation via step-level final-answer accuracy, physics-domain analysis, modality and reasoning-level comparisons, error analysis, and alignment with human expert judgments. Below is a concise summary of our key experimental findings:
> (1) Overall, frontier LLMs struggle to maintain coherent multi-step physics derivations despite strong performance on simpler tasks.  Model performance declines sharply with increasing problem difficulty
> (2)  Hard problems exhibit a sharp divergence between step-level and final-answer accuracy (final-answer accuracy falls below 10%), yielding extremely sparse rewards, while step-level scoring more faithfully captures partial reasoning competence and can serve as a dense signal for RL and data curation.
> (3)Domain analysis reveals consistent weaknesses in Quantum Mechanics, and most errors arise from incorrect physical assumptions (CAE), algebraic and computational mistakes (DCE) and Modeling & Process Understanding Errors(MPUE), providing clear guidance for future model training and improvement.
> (4) Multimodal input offers strong step-level gains for larger models but provides limited or even negative benefits for smaller models, where diagrams serve mainly a presentational role.
> (5) Deeper reasoning modes consistently enhance accuracy but require significantly more computation, highlighting a central efficiency challenge for scaling reasoning performance.
> (7) Our DAG-based, rule-driven evaluator aligns more closely with human experts than LLM-as-judge baselines.
> ```
>
> ### `W5 Clarification on step-level notation.`
> The step-level notation originates directly from the reference solutions provided in the data source. Based on these solutions, the DAG is constructed by LLM , which is then verified by rule-based and LLM-based checks. A detailed description of the full DAG construction and verification pipeline is provided in Section 5 of the paper.
>
> ## **Summary**
> We hope our responses sufficiently address your concerns regarding: (1) the clarity of Figures 1–5, (2) the reference for Fig1-2, (3) the alignment between challenges and methods, (4) the organization of the experimental summary, (5) the step-level notation.
> All significant changes made based on your feedback have been marked in red in the manuscript for your convenience.
>
> We sincerely thank the reviewer for recognizing the strengths of our work, including the curated DAG-structured benchmark, the scoring policy for fine-grained process evaluation, and the fully rule-based symbolic equivalence checker. We appreciate your reconsideration of our work and your thoughtful feedback.

---

> ### Author Response · Authors · 2025-11-28
>
> Dear reviewer,
>
> As the discussion period is nearing its end (with less than one week remaining), we would like to kindly confirm that we have fully addressed your concerns regarding:  (1) the clarity of Figures 1–5, (2) the reference for Fig1-2, (3) the alignment between challenges and methods, (4) the organization of the experimental summary, (5) the step-level notation, (6) Figures convey essential meaning more clearly, (7) The motivation clearly identifies gaps in existing physics datasets. If there are any additional points or feedback you would like us to consider, please let us know.
>
> We sincerely value your thoughtful comments and positive assessment of our DAG-structured benchmark, the interpretability of our DAG-based scoring policy, and the reliability of our rule-based symbolic equivalence checker.
>
> We appreciate that you have restored your score to its original level. If you now feel that your concerns have been fully addressed, we would be very grateful if you could consider increasing it.
>
> Best regards, Authors

---

### Official Review · Reviewer_PRSU · 2025-10-28

**Soundness:** 3
**Presentation:** 2
**Contribution:** 2
**Rating:** 4
**Confidence:** 4

**Summary:**

This paper introduces PRISM-PHYSICS, a framework for evaluating physics problems at the process level. Solutions are represented as directed acyclic graphs (DAGs) to capture dependencies between steps. The key innovation is the ancestor-closure scoring, which allocates partial credit based on intermediate steps. A rule-based symbolic equivalence checker ensures accurate comparison of formulas. Experimental results show that PRISM-PHYSICS provides detailed and reliable evaluations.

**Strengths:**

- PRISM-PHYSICS provides a large-scale, competition-level benchmark with carefully curated, DAG-structured solutions to complex physics problems.
- A fully rule-based symbolic equivalence checker ensures consistent validation of diverse mathematical expressions, eliminating reliance on heuristic LLM scoring and offering a more reliable comparison across alternative formulations.
- The ancestor-closure scoring policy allows for partial credit on intermediate steps, offering a more nuanced and fair assessment of student reasoning.

**Weaknesses:**

- Does the system account for context-dependent variations in formulas? (e.g., solving a problem from both kinematics and dynamics perspectives, or analyzing it through momentum and energy considerations)
- In physics problems, certain expressions may be contextually equivalent, but the strict analysis in this algorithm might overlook such context-dependent equivalence.   Does the current framework account for these context-sensitive variations?
- If skips over intermediate, simpler steps during the solution process, would this result in incorrect evaluation by the proposed method?
- The summary of the experiment is not clear enough. It is hoped that the author will have a more clear and organized discussion of the experimental results.

**Questions:**

See weakness.

---

> ### Author Response · Authors · 2025-11-20
>
> We sincerely thank the reviewer for recognizing the value of our large-scale benchmark, the rule-based symbolic equivalence checker, and the fairness of the ancestor-closure scoring policy. We address all concerns below and hope these clarifications will be helpful in your re-evaluation：
>
> ### ` W1 Context-Dependent Variations and Equivalent Formulations`
> We appreciate the reviewer’s insightful comment. We agree that physics problems often admit multiple valid solution paths (e.g., kinematics vs. dynamics, energy vs. momentum), and our framework is designed to accommodate such variation. The ancestor-closure scoring policy **does not require students to follow the same derivation as the reference solution**. As long as the student reaches a correct intermediate or final formula, the corresponding nodes in the reference DAG are matched and credit is assigned, **allowing alternative but valid reasoning paths** to receive partial or full credit.
>
> However, if a student’s approach produces formulas that align with none of the reference DAG nodes, the system lacks a verifiable anchor to determine whether the reasoning is physically sound. In this sense, the reference DAG acts as a lower bound on recognizable reasoning rather than an upper bound on what counts as valid reasoning. We do not penalize alternative approaches; we simply cannot assign credit when no step can be aligned with known physical relations. This limitation is inherent to any automatic evaluation system and is consistent with human grading, **when a derivation cannot be aligned with standard physics principles, even experts cannot confidently assign partial credit**.
>
> In practice, different solution methods often **converge to the same key intermediate equations**. For example, force analysis, conservation laws, and the Lagrangian formalism frequently lead to equivalent equations of motion in systems with holonomic constraints and conservative forces. Our DAG captures these shared nodes, and once the student reaches such a node, **back-propagation ensures that the necessary prerequisite steps receive appropriate credit**.
>
> Finally, most datasets provide only a single official solution per problem, and reliably constructing multiple alternative reasoning paths is non-trivial even for humans. Recent work such as LOCA [1] highlights the difficulty of rigorous reasoning-path augmentation. Given this limitation, we focus on constructing a single high-quality DAG per problem.
> This approach aligns with standard physics competition grading, where students using different but valid methods still receive partial credit once they reach key intermediate steps. And our human-annotation study in Section 6.5 confirms that this evaluation framework is also **the most consistent with human** scoring in practice.
>
> Reference:
>
> [1] LOCA: Logical Chain Augmentation for Scientific Corpus Cleaning, arXiv:2510.01249.

---

> > ### Author Response · Authors · 2025-11-20
> >
> > ### `W2 Context-Sensitive Equivalence and Robustness of the Matching Algorithm`
> >
> > We appreciate the reviewer’s insightful question, which was also a key consideration during the development of our framework.
> >
> > Indeed, the same physical relation can appear in multiple algebraic or conceptual forms, and **our framework is designed to capture a large portion of such context-dependent equivalence**. Our symbolic equivalence checker performs normalization such as variable renaming, algebraic simplification, and canonical restructuring, so expressions like
> > $$v^{2} = v_{0}^{2} + 2a(x - x_{0})\quad\text{and}\quad\frac{v^{2} - v_{0}^{2}}{2} = a(x - x_{0})$$
> > are treated as equivalent.
> >
> > We also experimented with explicitly substituting context-dependent equivalent expressions, but this approach quickly introduced ambiguity. In multi-case solutions, for example, formulas such as $v=v_0$​ and $v=2v_0$ may both appear to represent different regimes of the same system. **A naive context-aware matcher would incorrectly conflate these expressions.**
> > And **incorporating textual context to disambiguate such cases would require sophisticated natural-language understanding**, effectively **reintroducing an LLM into the evaluation loop** and contradicting our goal of a lightweight, deterministic framework.
> >
> > Moreover, these edge cases **occur extremely rarely** in our dataset, making the additional engineering complexity disproportionate to the marginal benefit.
> > Instead, we adopt a pragmatic and robust design choice: we require problem statements to explicitly specify all variables that should appear in the solution. This significantly **reduces the occurrence of ambiguous variable substitutions** while maintaining **a fully automatic, LLM-free evaluation pipeline**.
> >
> > Human annotation further shows that **our framework aligns most closely with expert judgments** among all compared methods. We acknowledge that developing a more advanced, fully context-aware symbolic matching algorithm that handles semantic equivalence without LLM assistance remains an open and technically challenging direction for future work.
> >
> > ### `W3 Handling of Skipped Intermediate Steps`
> > We appreciate the reviewer’s concern and would like to clarify that skipping simpler intermediate steps does not lead to incorrect evaluation. Under the ancestor-closure scoring policy, once a student reaches a correct intermediate or final formula, **all prerequisite nodes in the reference DAG are automatically marked as resolved**. This ensures that valid solutions which **omit trivial algebraic manipulations** or **merge multiple steps into a single expression** are not penalized. As long as the later steps are correct, the evaluation remains accurate and faithfully reflects the logical structure of the derivation.
> >
> > ### `W4 Clarity and Organization of Experimental Results`
> > Thank you for this valuable feedback. We have added a new subsection (6.2) that provides a concise one-sentence summary for each experiment to make the overall results clearer and more organized. We also put the content here for your reference.
> > ```
> > We conduct a comprehensive set of experiments, including overall performance evaluation via step-level final-answer accuracy, physics-domain analysis, modality and reasoning-level comparisons, error analysis, and alignment with human expert judgments. Below is a concise summary of our key experimental findings:
> > (1) Overall, frontier LLMs struggle to maintain coherent multi-step physics derivations despite strong performance on simpler tasks.  Model performance declines sharply with increasing problem difficulty
> > (2)  Hard problems exhibit a sharp divergence between step-level and final-answer accuracy (final-answer accuracy falls below 10%), yielding extremely sparse rewards, while step-level scoring more faithfully captures partial reasoning competence and can serve as a dense signal for RL and data curation.
> > (3) Domain analysis reveals consistent weaknesses in Quantum Mechanics, and most errors arise from incorrect physical assumptions (CAE), algebraic and computational mistakes (DCE) and Modeling & Process Understanding Errors(MPUE), providing clear guidance for future model training and improvement.
> > (4) Multimodal input offers strong step-level gains for larger models but provides limited or even negative benefits for smaller models, where diagrams serve mainly a presentational role.
> > (5) Deeper reasoning modes consistently enhance accuracy but require significantly more computation, highlighting a central efficiency challenge for scaling reasoning performance.
> > (6) Our DAG-based, rule-driven evaluator aligns more closely with human experts than LLM-as-judge baselines.
> > ```

---

> ### Author Response · Authors · 2025-11-20
>
> ## **Summary**
> We hope our responses sufficiently address your concerns regarding: (1) handling of context-dependent variations, (2) recognition of contextually equivalent expressions, (3) treatment of skipped intermediate steps, and (4) improved clarity for experiments. All significant changes made based on your feedback have been marked in red in the manuscript for your convenience.
>
> We sincerely thank the reviewer for recognizing the strengths of our large-scale benchmark, including the curated DAG-structured solutions, the fully rule-based symbolic equivalence checker, and the fair and nuanced ancestor-closure scoring policy. We truly appreciate your reconsideration of our work in light of our responses. Thank you again for your valuable and thoughtful feedback.

---

> ### Author Response · Authors · 2025-11-28
>
> Dear reviewer,
>
> As the discussion period is nearing its end (with less than one week remaining), we would like to kindly confirm that we have fully addressed your concerns regarding: (1) handling of context-dependent variations, (2) recognition of contextually equivalent expressions, (3) treatment of skipped intermediate steps, and (4) improved clarity for experiments. If there are any additional points or feedback you would like us to consider, please let us know.
>
> We sincerely appreciate your thoughtful comments and positive assessment of our large-scale benchmark, including the curated DAG-structured solutions, the fully rule-based symbolic equivalence checker, and the fair and nuanced ancestor-closure scoring policy. If you feel that your questions have been fully addressed, we would be very grateful if you could consider raising your score.  Thank you again for your valuable insights.
>
> Best regards, Authors

---

### Official Review · Reviewer_4ABk · 2025-10-30

**Soundness:** 2
**Presentation:** 2
**Contribution:** 2
**Rating:** 6
**Confidence:** 4

**Summary:**

This paper introduces PRISM-Physics, a large-scale physics reasoning benchmark with a proposed DAG-based evaluation protocol that addresses the limitations of existing LLM-as-judge scoring methods. The evaluation framework includes a fully rule-based symbolic formula equivalence checker to ensure consistent validation across diverse mathematical formulations, thereby eliminating reliance on subjective judgments. In the experiments, the paper investigates a diverse set of leading LLMs on PRISM-Physics and demonstrates the superiority of the proposed evaluation protocol compared to the LLM-as-judge method.

**Strengths:**

1. The idea of using DAG to judge the correctness of the final answer and intermediate steps is reasonable, and I agree that using the LLM-as-judge method to evaluate the correctness of physics problems is challenging and prone to errors.
2. The theoretical analysis part of the paper is solid.
3. The experiment is comprehensive and convincing.

**Weaknesses:**

1. My main concern is that, although PRISM-Physics can conduct rule-based judgments to determine the correctness of the final answer and intermediate steps using a DAG, the construction of the DAG still heavily relies on LLM-based extraction and rewriting. Compared to the existing LLM-as-judge method, the uncertainty introduced by LLMs seems to have merely shifted from the judgment stage to the preprocessing stage.
2. Another concern lies in the scalability and additional computational cost of the proposed evaluation protocol. Compared to existing benchmarks, PRISM-Physics requires an annotated DAG in addition to the final answer for each question in order to perform a more rigorous evaluation. Thus, the scalability of the proposed protocol appears limited. If we aim to extend this rigorous protocol to other existing benchmarks, what additional requirements would those questions need to meet? Furthermore, if we intend to construct a DAG for a new physics problem, how much extra computational cost would this introduce in the preprocessing stage?
3. Typo: in Line 362: "zero-shot **COTzheg** prompts". In Table 1, it would be better to retain the same number of digits after the decimal point and to bold the best results.
4. The results in Figure 8 (Appendix E.2) are difficult to understand. The authors should at least explain the meaning of each rectangle in the text and clarify whether the difference shown represents "multimodal – text" or "text – multimodal".

**Questions:**

See Weaknesses 1, 2.

---

> ### Author Response · Authors · 2025-11-20
>
> We sincerely thank the reviewer for recognizing the solid theoretical analysis, comprehensive experiments, and the reasonableness of our DAG-based evaluation approach. We appreciate the positive feedback and address the remaining concerns below.
> ### ` W1 Reliance on LLMs in DAG Construction and Potential Shift of Uncertainty`
> We appreciate the reviewer’s concern and would like to clarify why the DAG construction step does not simply shift uncertainty from evaluation to preprocessing:
>
> **(1) DAG construction is far simpler and more constrained than solution judging.**
>
> Unlike evaluating whether two full solutions are equivalent which is difficult even for human experts, the DAG only encodes explicit logical dependencies already present in the original solution. Most physics solutions **directly state relationships** such as *“The corresponding … are given by …”, “Substituting … into …, we get …”, “Setting …, we obtain …”*,  so the **LLM is not inferring structure from scratch but operating with strong textual and mathematical guidance rather than free-form speculation**. This makes the task substantially easier and keeps the error rate low.
>
> We further conducted **a human validation study on 70 problems** (10 from each domain), covering **912 formulas**. Human annotators found **no incorrect dependencies**, and **only about 1% of formulas exhibited minor, trivial dependency omissions.** These results indicate that LLM-based DAG construction is highly stable and introduces negligible uncertainty compared to the substantially harder task of end-to-end solution judging.
>
> **(2)  DAG construction is a one-time process and can be refined.**
>
> Since DAG construction is an offline preprocessing step, we can iteratively refine, review, and correct the DAGs until they are fully satisfactory. After this stage, the entire evaluation pipeline is rule-based and completely LLM-free, eliminating stochasticity at scoring time.
> In contrast, LLM-as-judge methods introduce uncertainty during evaluation, where such randomness cannot be removed or controlled.
>
> **(3) Rule-based matching and DAG scoring are independent components.**
>
> Our rule-based formula matching can also operate with a non-DAG scoring policy, which is entirely LLM-free and requires no DAG construction. We introduce the DAG-based scoring not out of necessity, but because it provides more principled partial credit for intermediate steps. These two contributions are logically independent.
>
> Taken together, the DAG construction step introduces minimal and controllable uncertainty, while enabling a strictly deterministic evaluation process that avoids the inherent randomness and inconsistency of LLM-as-judge scoring.
>
> ### ` W2 Scalability and Computational Cost of DAG Annotation and Extension to Other Benchmarks`
> We appreciate the reviewer’s insightful comment. The additional cost of our protocol is quite limited. As long as a dataset is provided in a required JSON format, our three-step rewriting pipeline can be applied directly without extra manual effort. DAG construction is only one component of this pipeline and requires a single LLM call per problem, making it lightweight compared to the overall cost of dataset creation.
> Moreover, DAGs are constructed once during preprocessing using the existing reference solution, which is far simpler than evaluating or generating full solutions. After this one-time step, the entire evaluation pipeline is fully rule-based and LLM-free, unlike LLM-as-judge methods that incur high computational cost and uncertainty at evaluation time.
> The protocol also imposes no additional requirements beyond having a reference solution in machine-readable form, making it straightforward to extend to other benchmarks.
>
> ### ` W3 Typo`
> Thank you for pointing this out. We have corrected the typo (“zero-shot COTzheg”) and updated Table 1 to use consistent decimal formatting and bold the best results in the revised PDF.
>
> ### ` W4  Clarification of Figure 8 Visualization and Difference Representation`
> Thank you for pointing out the confusion in Figure 8. We apologize for the unclear presentation. The values shown in the figure represent `multimodal – text` performance differences. Solid rectangles indicate positive gaps (multimodal outperforming text), while hatched rectangles indicate negative gaps. We have updated the caption and added a brief explanation in the text to make this clearer.
>
> ## **Summary**
> We hope our responses have clarified your concerns regarding: (1) the role of LLMs in DAG construction, (2) the scalability and cost of our evaluation protocol, (3) the corrected typo and formatting issues, and (4) the clarification of Figure 8. All significant changes made based on your feedback have been marked in red in the manuscript for your convenience.
>
> We appreciate the reviewer’s constructive feedback and thank you for reconsidering our work.

---

> ### Author Response · Authors · 2025-11-28
>
> Dear reviewer,
>
> As the discussion period is nearing its end (with less than one week remaining), we would like to kindly confirm that we have fully addressed your concerns regarding: (1) the role of LLMs in DAG construction, (2) the scalability and cost of our evaluation protocol, (3) the corrected typo and formatting issues, and (4) the clarification of Figure 8. If there are any additional points or feedback you would like us to consider, please let us know.
>
> We sincerely appreciate your thoughtful comments and positive assessment of the solid theoretical analysis, comprehensive experiments, and the reasonableness of our DAG-based evaluation approach. Thank you again for your valuable insights and hope you can consider raising your score.
>
> Best regards, Authors

---

### Official Review · Reviewer_Fpc6 · 2025-10-31

**Soundness:** 3
**Presentation:** 4
**Contribution:** 3
**Rating:** 6
**Confidence:** 3

**Summary:**

This paper introduces PRISM-Physics, a physics benchmark with particular focus on process-level evaluation. The authors model reasoning process as a Directed Acyclic Graph, where each node represents a formula in the resoning chain, and edges denote logical dependencies. They then propose to score a student answer by counting the ancestor closure of student's nodes within that of reference answer. Besides the scoring scheme, the authors established a way to reliably transform natural language reasoning into DAG with normalized formulas, allowing for robust formula matching during evaluation. Lastly, they curated PRISM-Physics, contributing to process-level evaluation of physics problem solving.

**Strengths:**

1. The paper is well written and very detailed.
2. Experiments and analysis are comprehensive and insightful, covering a wide range of LLMs and problem types.
3. The proposed DAG-based scoring scheme is novel and captures certain logical dependencies in the reasoning path.

**Weaknesses:**

1. The Ancestor Closure Scoring Policy seemed too forgiving for rigorous process-level evaluation. If I understand correctly, this scoring scheme overlooks skipped steps in the students reasoning path, as long as a targeted formula is attained downstream. Neither does this scheme check for validity of derivation, since it does not check whether assumptions leading to the formula are correct.
2. Following point 2, it would be nice to verify how well the scoring scheme aligns with true logical evaluation.
3. While the DAG-based scoring system does not require a student's answer to have the same sequential steps, it also ignores the structure in the student's answer. If I understand correctly, the student's answer is extracted as a bag of formulas with no logical dependencies extracted from context; only the reference answer is represented as DAG with logical dependency. Again, this seemed quite forgiving.

**Questions:**

1. Since you assume all ancestor nodes in the reference DAG are scored, have you checked how often a student answer actually misses those ancestor nodes? Would missing these nodes break the logical soundness of the student's answer?
2. In Section 6.4, did human experts score with Ancestor Closure Scoring as well? If yes, have you tried alternative evaluation methods (that is based on true logic instead of formula matching) to see how Ancestor Closure Scoring aligns with that scheme?
3. In Appendix E.1, did you use 8K context and zero temperature for evaluation of reasoning LLMs?
4. Have you tried extracting the student's answer as a DAG as well, and compare it to the reference DAG for a more rigorous scoring?

---

> ### Author Response · Authors · 2025-11-20
>
> We sincerely thank the reviewer for the positive and thoughtful feedback on improving our paper. We address all concerns below and hope these clarifications will be helpful in your re-evaluation：
> ### ` W1&W2&Q1 Concerns on Logical Soundness and Strictness of the Ancestor Closure Scoring Policy`
> We thank the reviewer for highlighting this important point. In physics, **one typically cannot arrive at a complex final equation through blind guessing**; it is impossible to write down a valid late-stage equation without consistent physical assumptions and coherent intermediate reasoning. Therefore, when a student (or model) reaches a correct downstream formula, **it inherently implies that the prerequisite physical relations have been correctly established**, even if some intermediate algebraic steps are not explicitly written. The Ancestor Closure Scoring Policy captures precisely this principle by rewarding logically necessary ancestors while still penalizing missing or inconsistent steps. Moreover, if a student follows **an alternative but physically valid reasoning path to derive the correct final formula**, it is appropriate to grant full credit, as the solution **remains sound from a physics standpoint and aligns with standard grading practices in physics competitions**.
>
> ### ` Q2 human Grading Standard`
> Thank you for pointing this out. No, the human experts graded based on their own professional judgment, as all annotators have strong physics competition backgrounds and are familiar with standard human grading practices. We have clarified this in the revised version.
>
> ### ` Q3 LLM Setting`
> Yes, we used an 8K context window and temperature 0 for all reasoning LLM evaluations.
>
> ### ` Q4 Extract Student Solution as DAG`
> We appreciate the reviewer’s insightful comment. The goal of our evaluation is to measure what percentage of the reference DAG is correctly achieved. Therefore, we only construct the reference DAG and check whether each student formula matches any node within it. **Since students may validly derive the answer through alternative reasoning paths, their solutions are not required to replicate the structure of the reference DAG, and constructing a student-side DAG is unnecessary for scoring.**
> A student-side DAG could be explored in future work to better characterize alternative reasoning strategies, but it is not necessary for our current scoring objective, which is defined solely in terms of reference-DAG coverage.
>
> ## **Summary**
> We hope our responses sufficiently address your concerns regarding: (1) ancestor scoring, (2) the human annotation setup, (3) the LLM inference config, and (4) extract student DAG. All significant changes made based on your feedback have been marked in red in the manuscript for your convenience.
>
> We sincerely appreciate your thoughtful comments and your positive assessment of our benchmark, scoring framework, and symbolic evaluation pipeline. Thank you again for your valuable insights.

---

> ### Author Response · Authors · 2025-11-28
>
> Dear reviewer,
>
> As the discussion period is nearing its end (with less than one week remaining), we would like to kindly confirm that we have fully addressed your concerns regarding (1) ancestor scoring, (2) the human annotation setup, (3) the LLM inference config, and (4) extract student DAG. If there are any additional points or feedback you would like us to consider, please let us know.
>
> We sincerely appreciate your thoughtful comments and positive assessment of our benchmark, scoring framework, and symbolic evaluation pipeline. Thank you again for your valuable insights and hope you can consider raising your score.
>
> Best regards, Authors

---

### Author Response · Authors · 2025-12-03

Dear ACs and Reviewers,
We sincerely thank you for the time and effort in evaluating our paper. To facilitate the next stage of discussion, we would like to (1) summarize our paper and our contribution; (2) the key points from the rebuttal as below:

### Key Strengths Highlighted by Reviewers
Olympiads have become useful testbeds for LLM performance. However, Physics Olympiads remain relatively underserved, especially that most recent physics benchmarks rely on evaluating final-answer only; most of these benchmarks rely on LLM-as-judge scoring; and some focusing on step-wise scoring utilizes straightforward linear step ordering, limiting the validity and generalization ability. Targeting these difficulties, we propose a benchmark constructing method, targeting a reliable evaluation of process scoring through: (1) we use the Direct Acyclic Graph-based process score calculation, and theoretically prove its optimality under certain constraints; and (2) use a fully rule-based symbolic formula equivalence checker for physics olympiads. Moreover, we (3) construct a large-scale benchmark of competition-level physics problems with these features (DAG and rule-based formula comparison), and (4) construct extensive experiments with wide range of state-of-the-art LLMs, both prove the validity of our proposed process-based scoring method, and provide an evaluation of current LLMs.

---

> ### Author Response · Authors · 2025-12-03
>
> ## **The key points from the rebuttal**
> Overall, most reviewers provide positive comments to our paper.
> ### **Key Strengths Highlighted by Reviewers**
>
> `Reviewer Fpc6` Experiments and analysis are **comprehensive and insightful**, The proposed DAG-based scoring scheme is **novel**
>
> `Reviewer 4ABk` The idea is reasonable, The theoretical analysis is **solid**, the experiment is **comprehensive and convincing**.
>
> `Reviewer PRSU` Large-scale benchmark  are **carefully curated**, fully rule-based symbolic formula equivalence checker  ensures **consistent validation**, The ancestor-closure scoring policy offering a **more nuanced and fair assessment.**
>
> `Reviewer 4yRy`  Large-scale benchmark  are **carefully curated**,  DAG enabling **fine-grained and interpretable** process-level evaluation, fully rule-based symbolic formula equivalence checker is **reliable**.

---

> > ### Author Response · Authors · 2025-12-03
> >
> > ### **How We Addressed Reviewer Concerns**
> > `Reviewer Fpc6`: We state that, (1) current LLMs and physics olympiad contestants are unlikely to arrive at correct final answer through blind guessing; and also (2) this kind of forgiving ancestor-scoring policy would be more robust for cases where the student is using a different method to arrive at the correct answer compared to ground truth answers. Moreover, in our paper where we compare DAG-based scoring and other scoring methods to human annotators, we are asking human annotators to come up with their own scoring mechanism and their own scores, so as to evaluate uncertainties in the scoring mechanism as well. In these experiments we show the superiority of our DAG-based scoring pipeline.
> >
> > `Reviewer 4ABk` :We state that, (1) unlike LLM-as-judge that has a high requirement of LLM abilities to compare formulas, DAG construction based on ground truth solutions are relatively simple and straightforward for LLMs. We **further conduct a human validation study** and found **only about 1%** formulas with wrong dependency generated in the LLM-based DAG construction, proving the robustness. Moreover, (2) DAG construction is a one-time process and can be refined Also, (3) rule-based physics formula scoring can also ensure more reliability compared to LLM-as-judges.We also fixed the typo and a better version of figure 8.
> >
> > `Reviewer PRSU`: We state that: **Regarding ancestor-closure scoring mechanism**, (1) we use the ancestor-closure mechanism so as to allow the students to receive credit to certain steps as long as the students find the result of those steps correct, which could accommodate the case where students are using correct derivations that deviate from ground truth in the process. Moreover, (2) this can ensure credits are given even if the student skips some intermediate steps (as long as later results are achieved). Furthermore, (3) experiments in Section 6.5 confirms the evaluation framework is the most consistent with human one among all scoring mechanisms we tested. **Regarding context-sensitive equivalence and robustness checking**, (1) our system can accommodate to certain cases such as different forms of the same equivalent equations, but for some extremely rare edge cases where there are some context-dependent equivalent expressions, to fully consider this would mostly require LLM-as-judge to label the contextual variables, which would introduce further uncertainty.
> >
> >
> > `Reviewer 4yRy` : **Regarding figure clarity**, we stated that we have tried our best to optimize readability of these figures through an enlargeable vector graph, and we further optimize it by including fewer models in the figure (and keeping detailed results in tables). **Regarding novelty**, we state the existing challenges of previous work: (1) the lack of fine-grained supervision of partial credit in existing benchmarks; (2) the non-linear, multi-branch nature of physics derivations of physics problems; and (3) the unreliability of LLM-as-judge approaches when scoring; and our corresponding novel solutions: (1) DAG-structured solutions to address the structural challenge and support principled partial credit, (2) Ancestor-Closure scoring that resolves the partial-credit challenge; and (3) the rule-based symbolic equivalence checker to replace LLM-as-judge checkers. To optimize presentation, we further modified the organization of the experimental summary.
> >
> > Once again, we sincerely thank the Area Chairs and Reviewers for their time and constructive feedback.
> >
> > With sincere respect and gratitude,
> >
> > Authors

---

### Meta-Review · Area_Chair_poHN · 2026-01-07

**Summary:**

The paper proposes a framework for evaluating physics problem solving at the process-level. This framework represents solutions in the form of directed acyclic graphs, where the nodes represent reasoning steps and edges represent their dependencies. Partial credit is assigned to solutions, based on intermediate steps that were derived correctly. This wasn't raised by the reviewers, but this is a paper with very similar findings for Math reasoning at ICML 2025: https://icml.cc/virtual/2025/poster/46462. Experiments with multiple LLMs show that PRISM-Physics provides more reliable and detailed evaluations of reasoning quality.

**Reviewer Concerns:**

Reviewers raised several concerns, but these were mostly handled by the authors during the rebuttals with clearly organized responses. I don't think any of these are left out.

**Reviewer Scores:**

I think all reviewers would preserve their scores.

---

### Decision · Program_Chairs · 2026-01-26

Accept (Poster)